

# Uncertainties identification of the blade-mounted lidar-based inflow wind speed measurements for robust feedback-feedforward control synthesis

Róbert Ungurán[1], Vlaho Petrović[1], Lucy Y. Pao[2], and Martin Kühn[1]

[1]ForWind, University of Oldenburg, Institute of Physics, Küpkersweg 70, 26129 Oldenburg, Germany
[2]Department of Electrical, Computer, and Energy Engineering, University of Colorado, Boulder, USA

*Correspondence to:* Róbert Ungurán (robert.unguran@uni-oldenburg.de)

**Abstract.** The current trend toward larger wind turbine rotors leads to high periodic loads across the components due to the non-uniformity of inflow across the rotor. On this regard, we introduce a blade-mounted lidar on each blade to provide a preview of inflow wind speed that can be used as a feedforward control input for the mitigation of such periodic blade loads. We present a method to easily determine blade-mounted lidar parameters, such as focus distance, telescope position, and orientation on the blade. However, such method is accompanied by uncertainties in the inflow wind speed measurement, which may also be due to the induction zone, wind evolution, "cyclops dilemma", unidentified misalignment in the telescope orientation, and the blade segment orientation sensor. Identification of these uncertainties allows their inclusion in the feedback–feedforward controller development for load mitigation. We perform large-eddy simulations, in which we simulate the blade-mounted lidar including the dynamic behaviour and the induction zone of one reference wind turbine for one above rated inflow wind speed. Our calculation approach provides a good trade-off between a fast-and-simple determination of the telescope parameters and an accurate inflow wind speed measurement. We identify and model the uncertainties, which then can directly be included in the feedback-feedforward controller design and analysis. The rotor induction effect increases the preview time, which needs to be considered in the controller development and implementation.

## 1  Introduction

The ongoing trend of steadily growing rotor diameter of wind turbines results in dynamic loads across the rotor swept area, which are becoming more uneven. Due to the so-called rotational sampling or eddy slicing effect, the blade samples the inhomogeneous wind field with frequencies determined by the rotor speed. Hence, the dynamic blade loads gets concentrated at the multiples of the rotational frequency, i.e., 1P, 2P, 3P,...,nP (Bossanyi (2003); van Engelen (2006)).

The scope of this paper is particularly geared to the relevance of three aspects of recent developments in controls to mitigate such loading. First, the control surfaces on the rotor are becoming more localized and consequently beside individual (blade) pitch control, local active or passive blade load mitigation concepts (e.g. trailing edge flaps) have been researched for several years. Second, in addition to the proven feedback control triggered by rotor speed or the individual blade root bending moments, feedforward control using either observer techniques or lidar-assisted preview information of the inflow is investigated



for collective or individual pitch as well as trailing edge flap control. Third, special attention is required in the feedback–feedforward controller design to guarantee robust stability and performance in the presence of inherent uncertainties in the lidar measurement.

The traditional collective pitch control (CPC) is responsible for keeping the rotor speed constant near and above rated wind
speed conditions. Bossanyi (2003) extended the CPC with individual pitch control (IPC) to mitigate the 1P dynamic blade load. The effectiveness of the IPC in reducing the dynamic blade loads is demonstrated in this paper. Later, the function of the IPC was extended to address the mitigation of higher harmonic dynamic blade loads (Bossanyi (2005); van Engelen (2006)), leading to load relief across the wind turbine components, i.e., blade root bending moments, hub yaw and tilt moments, yaw bearings, etc. Such a control design leads to the increased use of the blade pitch system. With growing blade length, the blade
mass rises with a power of two to three, and thus, increased pitch activity becomes even more undesirable, and as such results in wear and tear of the pitch actuators and bearings and equivalently, higher maintenance costs. One solution involves the use of small localized control surfaces to locally influence the thrust force, e.g., close to the blade tip, which contributes greatly to the overall blade root loadings. Pechlivanoglou (2013) conducted experimental and numerical studies to determine the most promising setup of passive and active local flow control solutions for wind turbine blades and he concluded that a controllable
flexible trailing edge flap close to the blade tip has the most potential to mitigate the dynamic blade loads. The individual trailing edge flap control (TEFC) is proven as an effective means of dynamic blade loads reductions in numerical studies (Bergami and Poulsen (2015); He et al. (2018); Ungurán and Kühn (2016); Zhang et al. (2018)), wind tunnel tests (Barlas et al. (2013); Marten et al. (2018); van Wingerden et al. (2011)), and field tests (Berg et al. (2014); Castaignet et al. (2014)). Castaignet et al. (2014) performed a full-scale test on the Vestas V27 wind turbine, reporting a load reduction of 14% at the
flap-wise blade root bending moment, providing proof of the control concept and the capabilities of the trailing edge flap for dynamic blade loads mitigation.

Recently, feedforward control has been identified as a promising concept for wind turbine control, as it mainly relies on indirect measurement of the disturbance, e.g., through measurement of rotor speed deviation from rated rotor speed or measurement of the blade root bending moment. Feedback controllers are only able to react on the disturbance after its influence
on the wind turbine has been measured, which leads to a delayed control action. Several authors propose lidar-assisted wind turbine controllers so that control actions can be determined before the disturbance influences the turbine. When properly tuned, this so-called feedforward control strategy can mitigate fatigue loading from external disturbances. The lidar-assisted collective pitch controller proposed by Schlipf et al. (2013) accomplished a better rotor speed tracking with reduced pitch activity, with respect to the feedback collective pitch controller. They demonstrated the reduction of damage equivalent loads
at the out-of-plane blade root bending moment, low-speed shaft torque, and tower bottom fore-aft bending moment through the use of lidar as feedforward collective pitch control input. Bossanyi et al. (2014); Kapp (2017) investigated the use of lidar for feedback–feedforward collective and individual pitch control and concluded its suitability for wind turbine control applications. Their purpose for the IPC was to mitigate the 1P loads at the flapwise blade root bending moment. They observed that a lidar-assisted feedback–feedforward IPC achieves marginal damage equivalent loads reduction with respect to feedback-only
IPC. Ungurán et al. (2019) achieved additional load reduction across various wind turbine components with the combined





feedback–feedforward IPC with respect to feedback-only IPC. They highlighted that to further reduce the blade root bending moment and avoid undesirable load increase on other wind turbine components, special care should be taken as the feedback is combined with feedforward IPC during controller development, in terms of, for instance, avoiding the same bandwidth for the feedback and feedforward IPC. This results in an elevated peak in the sensitivity function around the crossover frequency.

Furthermore, Bossanyi et al. (2014); Kapp (2017); Ungurán et al. (2019) studied different inflow wind conditions and wind turbine characteristics; they, also used different lidar systems for feedforward control purposes that influenced the results.

Due to obvious reasons, it is necessary to consider the uncertainties in the lidar measurements to achieve robust stability and performance of the feedback–feedforward controller. Furthermore, the source of such uncertainties must be identified and modeled, which can then be incorporated into the design and analysis of the controller, to ensure performance even for

uncertain lidar measurements. Several authors have already addressed this problem, e.g., Bossanyi (2013); Laks et al. (2013); Simley et al. (2014a, b) with their numerical investigations. Simley et al. (2016) performed field tests to assess the influence of "cyclops dilemma", spatial averaging error, induction zone, and wind evolution, on a hub-mounted lidar measurement. Simley et al. (2014a) used a hub-mounted continuous-wave (CW) lidar to investigate the effect of the "cyclops dilemma," and concluded the existence of a compromise in the preview distance. Spatial averaging increases with increasing distance from

the rotor plane, leading to correlation attenuation between the rotor-effective wind speed and the lidar-estimated inflow wind speed, with increasing frequency. As measurement gets closer to the rotor plane, the contribution of the lateral and vertical wind components to the line-of-sight lidar measurements also increases. Nevertheless, it is not possible to accurately reconstruct the longitudinal wind component from a single hub-mounted lidar system, which results in over- or underestimation of the rotor effective wind speed. Laks et al. (2013) investigated how wind evolution affects controller performance; they used a single point

measurement, without spatial averaging, in front of the wind turbine blade as a feedforward IPC input. Using the feedback–feedforward IPC, they acquired the highest load reduction at the blade root bending moment at a preview time of only 0.2 s. The further the measurement was taken from the rotor plane, the more the wind evolved on high frequencies (i.e., the so-called "wind evolution"), leading to overactuation by the feedforward IPC. It should be noted that the required preview time depends on many factors, e.g., wind turbine size, 1P frequency, inflow wind speed, induced phase shift by the feedforward controller

and blade pitch actuators, etc.

The blade-mounted lidar system is a novel technique that enables us to sample the wind component parallel to the rotor shaft axis around the swept area (Bossanyi (2013)) and has be demonstrated to be technologically viable (Mikkelsen et al. (2012)). Such a feature of the system enables addressing the mitigation of higher harmonic dynamic blade loads through feedback–feedforward individual pitch and trailing edge flap controllers (Ungurán et al. (2018, 2019)), while simultaneously posing

challenges with the presence of the induction zone. The closer the lidar measurement is taken to the rotor plane, the higher the deficit between the measured inflow and free flow wind speeds. Additionally, this deficit depends on where the lidar is mounted along the blade radius, which shows the importance of analyzing how the blade-mounted lidar measurement is affected by the wind evolution, the induction zone, and the assumptions made during the inflow wind speed reconstruction.





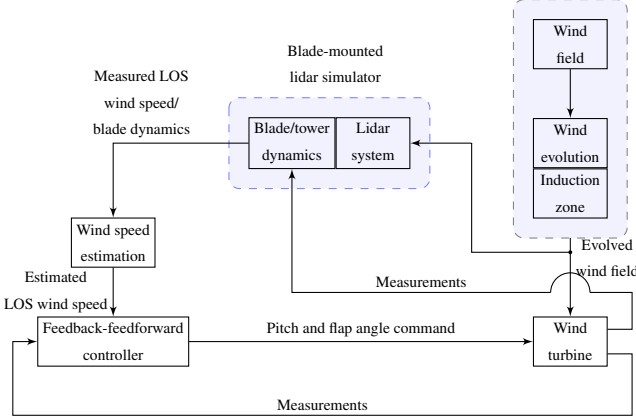

**Figure 1.** Block diagram of the blade-mounted lidar-based simulation setup. LOS corresponds to line-of-sight.

Therefore, in this study, our objective is to identify the uncertainties of the blade-mounted lidar measurement as a frequency-dependent uncertain weight for inclusion into the feedback–feedforward individual pitch and trailing edge flap control development, and to analyze the impact of the induction zone effect on the preview time.

The rest of the paper is organized as follows: Section 2 provides a description of the framework and methods we used for identifying the uncertainties and preview time of the blade-mounted lidar measurement, followed by an introduction of the blade-mounted lidar-based simulation setup in Section 2.1. The method we employed for determining the blade effective wind speed to assess the efficiency of the blade-mounted lidar-based inflow wind speed measurement is discussed in Section 2.2. Section 2.3 describes the general control implementation and presents the multiblade coordinate transformation and its importance in the controller design, while Section 2.4 details how the lidar-based measurement uncertainty in considered in control development and analysis. Section 2.5 proposes a method to identify the uncertainties of the blade-mounted lidar measurement as a frequency-dependent uncertainty weight, whereas Section 2.6 presents the method for the preview time estimation. The results of a reference case are presented in Section 3, with the establishment of the simulation setup (Section 3.1), an analysis on the effect of the multiblade coordinate transformation (Section 3.2), and a systematic analysis of the uncertainties of various telescope and control parameters (Section 3.3). Results of this paper are discussed in Section 4 prior to the conclusions in Section 5.

## 2 Methodology

### 2.1 Blade-mounted lidar

A telescope was mounted on each blade and was connected to a hub-based continuous-wave lidar with fiber optical cables. The lidar sampled the inflow wind speed in front of the rotor plane at a rate of 5 Hz, which we intend to use for control purposes. The lidar measurement was integrated into the system model according to Figure 1, through a combination of large-eddy





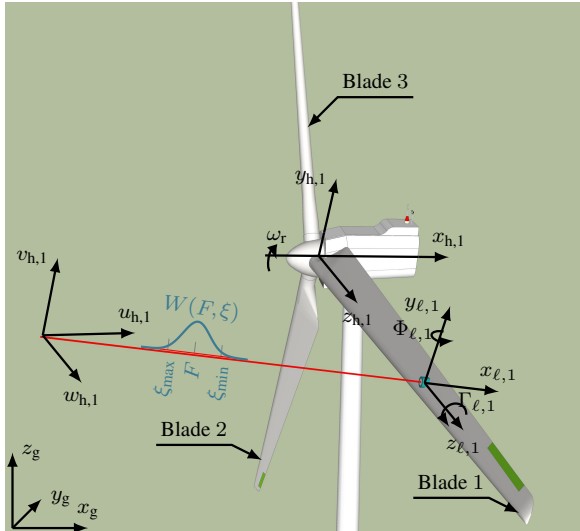

**Figure 2.** Configuration of the lidar measurement system, with a telescope mounted on each blade and connected to a continuous-wave lidar in the hub via fiber optics. The line-of-sight wind speed is computed on the basis of a weighting function ($W(F,\xi)$), which is dependent on the focus distance ($F$) and the range along the beam ($\xi$).

simulations and an aeroelastic simulation code. This enabled to simulate the lidar measurement in a realistic environment, where the effect of the induction zone and wind evolution, as well as the dynamic behavior of the wind turbine, were taken into account. Moreover, the lidar simulator considered volumetric measurement, dynamics of the blade and tower, i.e., displacement, rotation, and linear velocity in 3D space, and blade-rotation-induced velocity. Nevertheless, the rotational effect of the blade

5  was not accounted for during the accumulation of a single measurement.

Figure 2 illustrates the coordinate systems and the telescope orientation. Here, the line-of-sight (LOS) wind speed measurement from blade $i$ ($u_{\mathrm{los},i}$) could be defined as

$$u_{\mathrm{los},i} = \frac{\displaystyle\int_{\xi_{\min}}^{\xi_{\max}} W(F,\xi)\,V_i(\xi)\,d\xi}{\displaystyle\int_{\xi_{\min}}^{\xi_{\max}} W(F,\xi)\,d\xi}\;, \tag{1}$$

where $W(F,\xi)$ is the lidar's weighting function, defined according to Simley et al. (2014a) as

$$W(F,\xi) = \frac{1}{\xi^2 + \left(1 - \dfrac{\xi}{F}\right)^2 R_{\mathrm{R}}^2}\;, \tag{2}$$

where $R_{\mathrm{R}}$ is the Rayleigh range, set at 1,573 m herein, as proposed by Simley et al. (2014a); $F$ is the focus distance and $\xi$ is the range along the beam. Limits $\xi_{\min}$ and $\xi_{\max}$, introduced in Equation (1), refer to the minimum and maximum range,



respectively, along the beam. For practical implementation of the lidar simulator, these values were chosen such that $\frac{W(F,\xi)}{W(F,F)}$ equals 0.02. During discretization of Equation (1), the spatial resolution was set empirically at $\Delta\xi = 0.1\,\text{m}$. A single-point measurement is given by

$$V_i(\xi) \quad = \quad \left( \begin{bmatrix} u_{\text{h},i}(\xi) \\ v_{\text{h},i}(\xi) \\ w_{\text{h},i}(\xi) \end{bmatrix} - \begin{bmatrix} \dot{x}_{\text{t,h},i} \\ \dot{y}_{\text{t,h},i} \\ \dot{z}_{\text{t,h},i} \end{bmatrix} \right)^T \begin{bmatrix} \ell_{\text{x,h},i} \\ \ell_{\text{y,h},i} \\ \ell_{\text{z,h},i} \end{bmatrix} \quad , \tag{3}$$

where $[u_{\text{h},i}\ v_{\text{h},i}\ w_{\text{h},i}]^T$ is the wind speed vector along the laser beam expressed in the rotating hub coordinate system; $[\dot{x}_{\text{t,h},i}\ \dot{y}_{\text{t,h},i}\ \dot{z}_{\text{t,h},i}]^T$ is the linear velocity vector of the blade segment where the telescope is mounted, expressed in the rotating hub frame of reference and; $[\ell_{\text{x,h},i}\ \ell_{\text{y,h},i}\ \ell_{\text{z,h},i}]^T$ is the unit vector of the laser beam in the rotating hub coordinate system. The aeroelastic simulation tool is capable of providing full kinematics information, i.e., positions, orientations, and linear and angular velocities, of any blade segment in the hub coordinate system. During the inflow wind speed estimation, the velocity, displacement, and rotation

of the blade segment were assumed to be known; therefore, the measured LOS wind speed can be corrected as indicated in Equation (4).

Without loss of generality, two assumptions were made: (1) the $v_{\text{h},i}$ and $w_{\text{h},i}$ components are zero and (2) the mean wind speed is parallel with the rotor axis, i.e., no tilt and no yaw misalignment is considered. Consequently, the wind speed parallel to the rotor shaft axis ($u_{\text{h,est},i}$) can be estimated as

$$u_{\text{h,est},i} \quad \approx \quad \frac{u_{\text{los},i} + \dot{y}_{\text{t,h},i}\ell_{\text{y,h},i} + \dot{z}_{\text{t,h},i}\ell_{\text{z,h},i}}{\ell_{\text{x,h},i}} + \dot{x}_{\text{t,h},i} \quad . \tag{4}$$

Nevertheless, such assumptions introduced errors in the lidar measurement that were presumed to exist in the identified uncertainty weight, and thus, were consequently considered during the controller development.

## 2.2   Blade effective wind speed and wind speed deficit estimation

To assess the performance efficiency of the blade-mounted lidar-based inflow wind speed measurement, we introduced a new

signal called the blade-effective wind speed ($u_{\text{beff},i}$), which is determined as the contribution of the inflow wind speed on each blade segment $u_i(r)$ to the flapwise blade root bending moment; the inflow wind speed refers to the longitudinal wind speed in the rotor axis direction. The contribution depends on the radial distance ($r$) and the local thrust coefficient ($C_{\text{T}}$) of the blade segment as expressed by

$$u_{\text{beff},i} \quad = \quad \sqrt{\frac{\displaystyle\int_{R_{\text{hub}}}^{R_{\text{tip}}} C_{\text{T}}(r, u_i(r))\, r^2\, u_i^2(r)\, dr}{\displaystyle\int_{R_{\text{hub}}}^{R_{\text{tip}}} C_{\text{T}}(r, u_i(r))\, r^2\, dr}} \quad . \tag{5}$$

The local thrust coefficients were resolved from steady-state simulations for each blade segment from cut-in to cut-out wind speeds.





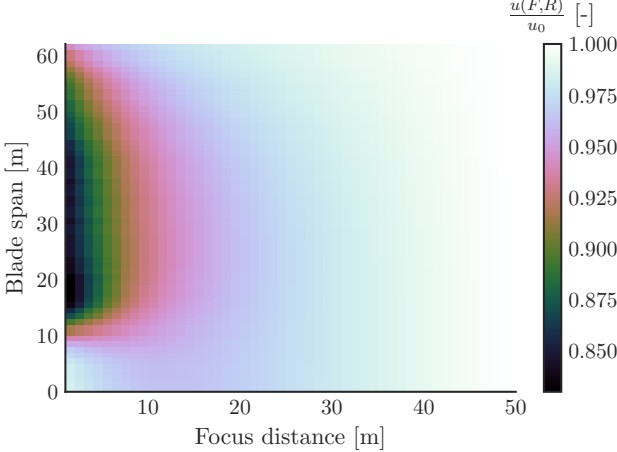

**Figure 3.** Normalized longitudinal inflow wind speed ($\frac{u(F,R)}{u_0}$) for different focus distance ($F$) and blade span position ($R$), with an undisturbed inflow wind speed $u_0 = 13\,\mathrm{m\,s^{-1}}$.

Figure 3 illustrates the induction zone effect for the reference case defined in Section 3.1. Note that the lidar measurement was affected by the rotor induction. The reduction depends on the position of the telescope along the blade radius ($R$) and the focus distance of the laser beam ($F$), where the wind speed measurement takes place. To account for this effect in the lidar-based inflow wind speed measurement, we constructed a second-order polynomial function, whose inputs were chosen from

widely available wind turbine sensors, such as rotor speed ($\omega_\mathrm{r}$), blade pitch angle ($\beta_i$), and blade root flapwise and edgewise moments ($M_{\mathrm{fw},i}$, $M_{\mathrm{ew},i}$). Therefore, the estimated wind speed parallel to the rotor shaft axis ($u_{\mathrm{h,est},i}$) is corrected as

$$u_{\mathrm{cor},i} \quad = \quad u_{\mathrm{h,est},i} + \Delta u_{\mathrm{est},i} \ , \tag{6}$$

where

$$u_0 - u(F,R) \approx \Delta u_{\mathrm{est},i} = f(F,R,\omega_\mathrm{r},\beta_i,M_{\mathrm{fw},i},M_{\mathrm{ew},i}) \ . \tag{7}$$

**2.3   Multiblade coordinate transformation (MBC)**

In the subsequent step, we introduced the multiblade coordinate transformation (MBC) that simplifies the controller design by transforming a time-varying system into a time-invariant system and decouples the individual pitch from the collective pitch control. Figure 4 demonstrates the manner in which the feedforward controller was implemented. First, the measured inflow wind speed was transformed to the non-rotating frame of reference by applying MBC transformation ($T_{\mathrm{mbc}}(\theta + \phi)$) in

accordance with Equation (8), where $\theta$ denotes the azimuth angle.

$$\begin{bmatrix} u_{\mathrm{cor,col}} \\ u_{\mathrm{cor,yaw}} \\ u_{\mathrm{cor,tilt}} \end{bmatrix} \quad = \quad T_{\mathrm{mbc}}(\theta) \begin{bmatrix} u_{\mathrm{cor,1}} \\ u_{\mathrm{cor,2}} \\ u_{\mathrm{cor,3}} \end{bmatrix} \tag{8}$$



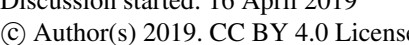

**Figure 4.** Implementation of the feedforward collective and individual pitch control, where the inputs ($u_{\text{cor},1}$, $u_{\text{cor},2}$, and $u_{\text{cor},3}$) are the estimated wind speeds parallel to the rotor shaft axis and the outputs are the blade pitch angles ($\beta_1$, $\beta_2$, and $\beta_3$). The feedforward controller ($K_{\text{ff,f}}$) was implemented in the non-rotating (fixed) frame of the reference and was, therefore, denoted with an extra index f. Further, the multi-blade coordinate transformation ($T_{\text{mbc}}$) was applied to the inputs, and the pseudo-inverse transformation ($T_{\text{mbc}}^+$) was applied to the outputs.

where

$$
T_{\text{mbc}}(\theta) = \begin{bmatrix} \frac{1}{3} & \frac{1}{3} & \frac{1}{3} \\ \frac{2}{3}\cos(n_{\text{h}}\theta) & \frac{2}{3}\cos(n_{\text{h}}[\theta+\frac{2\pi}{3}]) & \frac{2}{3}\cos(n_{\text{h}}[\theta+\frac{4\pi}{3}]) \\ \frac{2}{3}\sin(n_{\text{h}}\theta) & \frac{2}{3}\sin(n_{\text{h}}[\theta+\frac{2\pi}{3}]) & \frac{2}{3}\sin(n_{\text{h}}[\theta+\frac{4\pi}{3}]) \end{bmatrix}. \tag{9}
$$

A phase shift ($\phi$) was introduced into the transformation to consider that the measured inflow wind speed hits the wind turbine blade after this azimuth angle change. This value varies with respect to several parameters, including the selected focus distance, inflow wind speed, and rotor speed. Further, the control signals or the blade pitch angles ($\beta_{\text{col}}$, $\beta_{\text{yaw}}$, $\beta_{\text{tilt}}$) were determined by the feedforward controller ($K_{\text{ff,f}}$). If the preview time provided by the lidar was higher than the time delay induced by the feedforward controller, an additional time delay ($e^{-sT_{\text{id}}}$) was introduced into the system. Finally, the delayed control signals ($\beta_{\text{col,d}}$, $\beta_{\text{yaw,d}}$, and $\beta_{\text{tilt,d}}$) were transformed to the rotating frame of the reference using the pseudo-inverse MBC transformation ($T_{\text{mbc}}^+(\theta)$). The main structure of the feedforward individual pitch controller in Figure 4 can be used in the feedforward trailing edge flap controller as well.

The MBC transformation plays a considerably important role because it can transform a frequency component of interest, such as 1P, 2P, or 3P (Bossanyi (2003); van Engelen (2006)), to a low-frequency component, named as 0P. It is dependent on the selected value of $n_{\text{h}}$ in Equation (8). For example, 1P will be transformed to 0P when $n_{\text{h}}$ is specified as 1, and 2P will be transformed to 0P when $n_{\text{h}}$ is specified as 2.

In this study, we focus on identifying the uncertainty weight that can be used during the feedback-feedforward individual and collective pitch control development with an objective to mitigate the 1P loads at the flapwise blade root bending moments and to enhance the rotor speed tracking. This indicates that the measured inflow wind speeds were transformed to the non-rotating frame of reference by considering $n_{\text{h}}$ as 1 in Equation (8), where the uncertainty weight identification was conducted. Further, the same methodology can be applied to identify the uncertainty weight for high harmonics control by selecting a large integer value of $n_{\text{h}}$.

## 2.4 System modeling with uncertain lidar measurements

We used the blade-mounted telescopes to measure the disturbance, or the inflow wind speed in this case. Afterward, the three measurements were transformed into the non-rotating frame of reference where they were used as inputs to the feedforward individual and collective pitch controllers. Figure 5 illustrates the disturbance rejection controller setup with uncertainty. Each




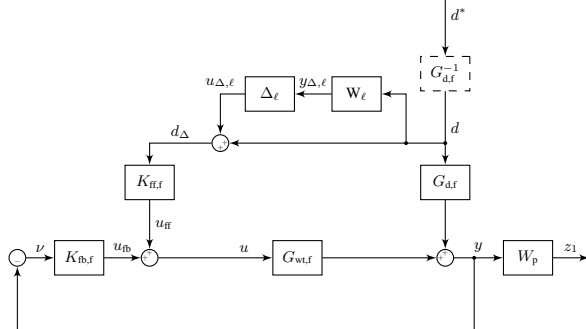

**Figure 5.** Block diagram of the disturbance rejection control design with performance weight and uncertain input measurement. $K_{\text{fb,f}}$, $K_{\text{ff,f}}$ are the feedback and feedforward controllers, $G_{\text{wt,f}}$ is the wind turbine model, $G_{\text{d,f}}$ is the disturbance model, $\Delta_\ell$ is the uncertainty, $W_\ell$ is the uncertainty weight, and $W_{\text{p}}$ is the performance weight. The f in the index refers to the non-rotating (fixed) frame of reference.

block in the figure represents a three-input and three-output system. Consequently, the resulting transfer function was in a $3 \times 3$ matrix (three-input and three-output). The measurement uncertainty can vary with wind speed, wind shear, turbulence intensity, etc. (Navalkar et al. (2015)), thus, multiplicative diagonal complex uncertainties were considered.

The control development was aimed at achieving disturbance rejection up to a certain frequency with measurement un-
5   certainties. In other words, we wanted to find a controller that satisfies Equation (10) for a chosen performance weight $W_{\text{p}}$.

$$\left\| \, W_{\text{p}} \, S_{\text{fb}} \, S_{\text{ff,p}} \, \right\|_\infty < 1, \tag{10}$$

where the frequency-dependent feedback ($S_{\text{fb}}$) and feedforward sensitivity ($S_{\text{ff,p}}$) functions with multiplicative uncertainty are given by

$$\begin{aligned} S_{\text{fb}} &= (I + G_{\text{wt,f}} K_{\text{fb,f}})^{-1} \\ S_{\text{ff,p}} &= I + G_{\text{wt,f}} K_{\text{ff,f}} (I + \Delta_\ell W_\ell) \, G_{\text{d,f}}^{-1} \end{aligned} \tag{11}$$

and

$$\Delta_\ell = \begin{bmatrix} \delta_{\ell,1} & 0 & 0 \\ 0 & \delta_{\ell,2} & 0 \\ 0 & 0 & \delta_{\ell,3} \end{bmatrix} \in \mathbb{C}^{3 \times 3} \, , \tag{12}$$

for property $\|\Delta_\ell\|_\infty \leq 1$. This equation highlights the importance of knowing the frequency-dependent uncertainty weight $W_\ell(j\,\omega)$ in advance, so as to ensure that the closed-loop system is stable and that the objective in Equation (10) is satisfied for
15   all perturbations ($\|\Delta_\ell\|_\infty \leq 1$). For control development, only the identification of the frequency dependent weight of $W_\ell(j\,\omega)$ was missing, which was identified for the reference case in Section 3.3.

*Remarks:* (1) Only one objective was introduced in Equation (10); nevertheless, other objectives can be added, such as penalizing the control signal magnitude at high frequencies (Ungurán et al. (2019)). (2) To avoid the disturbance model acting





as a scaling factor of the objective function, as in

$$z_1 = W_{\mathrm{p}} \, S_{\mathrm{fb}} \, S_{\mathrm{ff,p}} \, G_{\mathrm{d,f}} \, d, \tag{13}$$

Figure 5 was extended with the inverse of the disturbance model ($G_{\mathrm{d,f}}^{-1}$) (shown in a dashed rectangle), so that

$$z_1 = W_{\mathrm{p}} \, S_{\mathrm{fb}} \, S_{\mathrm{ff,p}} \, d^* \quad , \tag{14}$$

5    which ensures that $z_1$ is not affected by the disturbance model. Hence, in the control synthesis and analysis, $z_1$ is a direct indicator of the controller performance in the presence of uncertainties.

## 2.5   Uncertainty modeling for control development

We employed black box system identification to establish the transfer functions ($G_\ell$) from the blade effective wind speeds ($u_{\mathrm{beff}}$) to the corrected lidar based inflow wind speeds ($u_{\mathrm{cor}}$) in the non-rotating (fixed) frame of reference

$$u_{\mathrm{cor,f}} \;=\; G_\ell \, u_{\mathrm{beff,f}} \tag{15}$$

with

$$G_\ell \;=\; \begin{bmatrix} G_{\ell,\mathrm{col}} & 0 & 0 \\ 0 & G_{\ell,\mathrm{yaw}} & 0 \\ 0 & 0 & G_{\ell,\mathrm{tilt}} \end{bmatrix} \in \mathbb{C}^{3\times 3} \quad . \tag{16}$$

The system identification is performed via the `ssest` function from MATLAB (2018) with a 15th-order state-space model, which can capture all the relevant information. The order of the state-space model was found empirically.

15    We separately identified the uncertainty weight for each of the inputs ($w_{\ell,k}(j\,\omega)$) in such a way as to ensure that the relative error between the nominal ($G_{\mathrm{n},k}(j\,\omega)$) and the identified systems ($G_{\ell,k}(j\,\omega)$) is below each uncertainty weight

$$\left| \frac{G_{\mathrm{n},k}(j\,\omega) - G_{\ell,k}(j\,\omega)}{G_{\mathrm{n},k}(j\,\omega)} \right| \;<\; |w_{\ell,k}(j\,\omega)|, \; \forall \omega \quad . \tag{17}$$

The uncertainty weight is modeled as a first-order minimum-phase filter

$$w_{\ell,k}(j\,\omega) \;=\; w_{\mathrm{DC},k} \, \frac{1 + j\,\omega \, \frac{w_{\infty,k}}{w_{\mathrm{DC},k}} \, T}{1 + j\,\omega\, T} \tag{18}$$

20   where

$$T \;=\; \frac{1}{\omega_{0,k} \sqrt{\frac{w_{\infty,k}^2 - 1}{1 - w_{\mathrm{DC},k}^2}}} \, , \quad . \tag{19}$$

Here, $w_{\mathrm{DC},k} = w_{\ell,k}(j\,0)$ and $w_{\infty,k} = w_{\ell,k}(j\,\infty)$ represent the DC and high-frequency gains of the filter, and correspond to the uncertainties at low and high frequencies, respectively. The crossover frequency $\omega_{0,k}$ is defined as the frequency where



the magnitude of the filter crosses 1 from below ($|w_{\ell,k}(j\,\omega_{0,k})| = 1$), or 0 dB, and with $k \in \{\text{col, yaw, tilt}\}$, leading to the frequency-dependent diagonal weighting matrix of

$$
W_\ell = \begin{bmatrix} w_{\ell,\text{col}} & 0 & 0 \\ 0 & w_{\ell,\text{yaw}} & 0 \\ 0 & 0 & w_{\ell,\text{tilt}} \end{bmatrix} , \tag{20}
$$

which can be used in the feedback–feedforward IPC control development and analysis. The expressions $w_{\text{DC},k}$, $w_{\infty,k}$, and $\omega_{0,k}$

are identified for several cases in Section 3.3.

The ideal case would be to measure with a telescope, the exact inflow wind speed hitting the rotor blades, to result in a transfer function with a gain of 1 over the entire frequency range. Therefore, we chose a first-order Butterworth low-pass filter with a cut-off frequency of 10 Hz and a gain of 1, as the nominal system ($G_{\text{n},k}$). With the lidar having a sampling rate of 5 Hz, we ensured that the gain up to 2.5 Hz was close to 1. Higher frequencies were not studied in this work. We neglected the

cross-coupling between the yaw and tilt components in the system identification, but these were considered in the wind turbine and disturbance transfer functions in line with Lu et al. (2015), so that the cross-coupling between the yaw and tilt components is included in the controller development.

### 2.6 Preview time estimation

Preview time plays an important role in the development of feedforward control. It must be larger than or equal to the time delay

introduced by the feedforward controller and actuator dynamics. It is preferable to be equal, but a larger value is acceptable, as additional time delay can be easily introduced into the feedforward controller, as shown in Figure 4. To determine the optimal preview time for a given focus distance, we built three functions ($J_k$) based on the coherence ($\gamma_k^2$) and phase shift ($\varphi_k$) between the blade effective ($u_{\text{beff},k}$) and the corrected inflow ($u_{\text{cor},k}$) wind speeds, with $k \in \{\text{col, yaw, tilt}\}$. The power spectral density ($S_k$) of the blade effective wind speeds gives more weight to the relevant frequencies where power is concentrated. The final

function $J$ is the sum of the three functions $J_k$

$$
J \;=\; \sum_k J_k \;=\; \sum_k \frac{\displaystyle\int_0^{f_{\max}} S_k(f)\frac{|\varphi_k(f)|}{\gamma_k^2(f)}\,df}{\displaystyle\int_0^{f_{\max}} S_k(f)\,df} \;. \tag{21}
$$

A near-optimal preview time is obtained by delaying the corrected inflow wind speed measurement through an assumption of a preview time range and evaluation of Equation (21) for each delayed case.



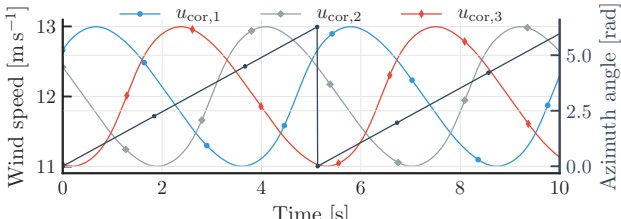

**Figure 6.** Time series of three generic wind speed measurements at the same amplitude, used for analyzing the impact of the multiblade coordinate transformation. The first, second, and third signals have a phase shift of $30°$, $150°$, and $270°$, respectively. The signals are constructed to include harmonics up to 6P.

## 3 Result

### 3.1 Simulation setup

The reference case we used in this investigation was based on the NREL 5 MW generic wind turbine (Jonkman et al. (2009)). We used an actuator line model through the coupling between FASTv7 (Fatigue, Aerodynamics, Structures, and Turbulence)

aeroelastic simulation code (Jonkman and Buhl (2005)) and PALM (Parallelized Large-Eddy Simulation Model) (Maronga et al. (2015)) as explained by Bromm et al. (2017). The operating conditions corresponded to a resulting hub-height mean wind speed of $13.06 \, \mathrm{m \, s^{-1}}$, which is above the rated value of $11.4 \, \mathrm{m \, s^{-1}}$. Furthermore, the simulation resulted in a turbulence intensity of $8.5\,\%$, and a wind shear corresponding to a power law description with an exponent of approximate 0.12. The baseline controller of the wind turbine ensured that the generator speed is kept at 1173.7 rpm (Jonkman et al. (2009)), thereby

resulting in a mean rotor speed ($\omega_\mathrm{r}$) of 11.74 rpm and further leading to a 1P frequency of $f_0 = 0.195 \, \mathrm{Hz}$.

For an analysis of the induction zone effect, we set the range of the focus distance and telescope position along the blade radius at $F \in [10\,\mathrm{m}, 40\,\mathrm{m}]$, $R \in [20\,\mathrm{m}, and \, 60\,\mathrm{m}]$, based on a previous investigation (Ungurán et al. (2018)). The range of the other input variables were determined by the results for the simulations with laminar inflow and power law wind shear with coefficients of 0.1, 0.2, and 0.3. An approximation of the induction zone effect introduced some uncertainties into the measurement,

but they were included in the identified uncertainty weight.

### 3.2 Multiblade coordinate transformation effect on the blade-mounted lidar measurement

To perform an analysis of the MBC transformation, we created three generic wind speed measurement signals with

$$u_{\mathrm{cor},i} = u_0 + \sum_{j=1}^{6} \frac{1}{j^3} \sin\left( j \left[ 2\pi f_0 t + (i-1)\frac{2\pi}{3} + \frac{\pi}{6} \right] \right), \tag{22}$$

where $u_0$, $i$, $f_0$, and $t$ are the offset, blade index, 1P frequency, and time, respectively. Here, we considered harmonics of

up to 6P ($j = 1 \dots 6$). Figure 6 shows a sample time series of the generated signals. Figure 7 presents the power spectral density of the wind speed measurement obtained from the first blade ($u_{\mathrm{cor},1}$) and the collective ($u_{\mathrm{col}}$), yaw ($u_{\mathrm{yaw}}$), and tilt ($u_{\mathrm{tilt}}$)



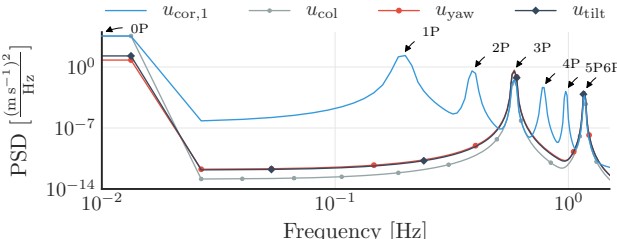

**Figure 7.** Power spectral density of the generic signals in the rotating ($u_{\text{cor},1}$) and non-rotating ($u_{\text{col}}$, $u_{\text{yaw}}$, $u_{\text{tilt}}$) frames of reference during the application of the multiblade coordinate transformation.

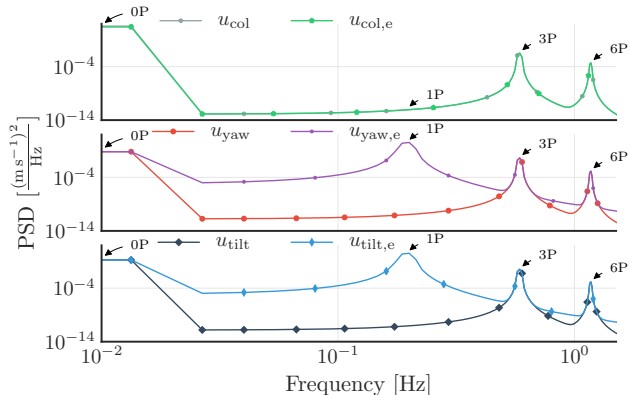

**Figure 8.** Power spectral density of collective, yaw, and tilt components of the generic signals with partial DC offset. The expression u$_{...,e}$ indicates the case where the DC offset ($u_0$ in Equation (22)) of one of the signals differs from the other two in the rotating frame of reference.

components after the MBC transformation, which was applied on the generic wind speed measurement signals ($u_{\text{cor},1}$, $u_{\text{cor},2}$, $u_{\text{cor},3}$). The figure highlighted the MBC transformation keeping only 0P, 3P, and multiples of 3P. As Lu et al. (2015) described, the frequency ($f$) in the non-rotating frame of reference arises from $f \pm f_0$ from the rotating frame of reference, e.g., the 3P in the non-rotating frame of reference arises from the 2P and 4P contributions in the rotating frame of reference.

5    Several cases may illustrate the transfer of the measurement errors from the rotating to the non-rotating reference frame. First, we should consider the effect of over- or underestimation of the measured wind speed with one of the blade-mounted lidar systems, due to e.g., different radial positions of the telescope along the blade radii or one of the telescopes having a different orientation, which reduced the DC offset ($u_0$ in Equation (22)) for one of the three generic signals. Next, the signals were transformed into the non-rotating frame of reference, which can be compared to the case where all the DC offsets were

10    maintained for each of the three signals at the same level. As Figure 8 highlights, an undesired peak appeared at 1P in the yaw and tilt components in the non-rotating frame of reference, due to the presence of asymmetries in the signals in the rotating frame of reference (Petrović et al. (2015)).





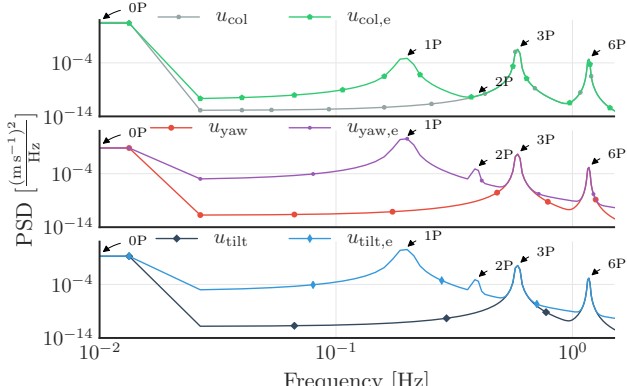

**Figure 9.** Power spectral density of collective, yaw, and tilt components of the generic signals with partial DC offset and phase shift. The expression u...,e indicates the case where a different DC offset is set and a $1°$ of phase shift is added to the 1P harmonics of one of the blade signals in the rotating frame of reference.

Second, aside from the reduction of the DC offset for one of the signals, a $1°$ of phase shift was added to the 1P harmonics in the rotating frame of reference, which represents the case, for example, where one of the blade-mounted lidar focus distances differs from the other two. Figure 9 reveals that after applying the MBC transformation to the three generic signals, undesired higher harmonic peaks rose in the non-rotating frame of reference. Interestingly, the phase shift that was introduced to one of
the signals in the rotating frame of reference resulted to different higher harmonics in the components in the non-rotating frame of reference, e.g., a peak observed at 1P of the collective component and at 2P of the tilt and yaw components.

### 3.3  Uncertainty weight identification

Ungurán et al. (2019) stressed that an elevated peak around the crossover frequency (just below the 1P frequency) of the feedback–feedforward controller sensitivity function leads to increased loads across the wind turbine components. Here, the
crossover frequency of the controller was defined where the sensitivity function first crosses the -3 dB from below. Uncertainties pose limitations on the achievable performance (Skogestad and Postlethwaite (2005)), e.g., the peak of the sensitivity function may increase due to uncertainties in the system. Therefore, it is important to analyze how the lidar measurement uncertainty is affected by e.g., mounting misalignment of the telescope on the blade, or in cases where the focus distance or position of the telescope along the blade span differs from the optimal parameters, etc. Identifying the lidar measurement uncertainty as
a frequency-dependent first-order minimum-phase filter enables the inclusion of such parameters in the control development, allowing an analysis of its impact on the stability and performance of the closed-loop system. A straightforward solution to determine the telescope and lidar parameters, such as focus distance, telescope position along the blade radius, telescope orientation on the blade, etc., is to assume that the blades are rigid, rotor speed and pitch angle are constant, and that Taylor's frozen turbulence hypothesis (Taylor (1938)) holds (Ungurán et al. (2018)). We performed large-eddy simulation (LES) in the





**Table 1.** The cases investigated in this study, along with the lidar and telescope parameters for each case. If one or more parameters in the third column are not specified, then the parameters defined in the first case are used. $F$ is the focus length, $R$ is the radial position of the telescope along the blade, and $\Phi_{\ell,i}$ and $\Gamma_{\ell,i}$ are the orientation angles of the telescope.

| Case | Uncertainties for: | Parameters |
|---|---|---|
| $C_1$ | telescope parameters from literatures, assuming: <br> – no induction <br> – no wind evolution <br> – no blade flexibility | $F = 22.2\,\mathrm{m}$ <br> $R = 44\,\mathrm{m}$ <br> $\Phi_{\ell,i} = -3.7°$ <br> $\Gamma_{\ell,i} = 7.0°$ |
| $C_2$ | telescope parameters within prescribed range | $F \in [20.2\,\mathrm{m}, 30\,\mathrm{m}]$ <br> $R \in [43\,\mathrm{m}, 45\,\mathrm{m}]$ <br> $\Phi_{\ell,i} \in [-5.7°, -1.7°]$ <br> $\Gamma_{\ell,i} \in [5°, 9°]$ |
| $C_3$ | different telescope focus length | $F \in [20.2\,\mathrm{m}, 30\,\mathrm{m}]$ |
| $C_4$ | different position of the telescope along the blade radius | $R \in [42\,\mathrm{m}, 46\,\mathrm{m}]$ |
| $C_5$ | different orientation angles of the telescope | $\Phi_{\ell,i} \in [-6.7°, -0.7°]$ <br> $\Gamma_{\ell,i} \in [4°, 10°]$ |



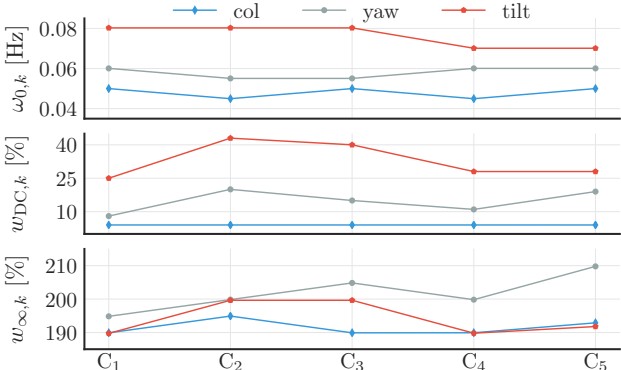

**Figure 10.** Identified parameters of the first-order minimum-phase filter. The expression $\omega_{0,k}$ is the crossover frequency of the filter; $w_{\mathrm{DC},k}$ and $w_{\infty,k}$ represent the DC and high-frequency gains of the filter expressed as percentages, with $k \in \{\mathrm{col, yaw, tilt}\}$; $\mathrm{C_1, C_2, C_3, C_4}$, and $\mathrm{C_5}$ represent the investigated cases (as discussed in the subsequent subsections).

succeeding sections to examine the usefulness and limitations of these assumptions, and further analyzed the uncertainties in the blade-mounted lidar measurement as well as the measurement sensitivity with respect to lidar and telescope parameter changes. The investigated cases are described in Sections 3.3.1 to 3.3.5, and summarized in Table 1. Section 3.3.6 describes how the measurement uncertainties are affected when one or two telescopes are aligned differently than the others. First, we

assumed that the orientation angle misalignment was unknown. Second, we assumed that this orientation angle misalignment can be identified, so that the lidar-based inflow wind speed measurement can be corrected.were

For each case, the relative error between the nominal ($G_{\mathrm{n},k}(j\,\omega)$) and the identified ($G_{\ell,k}(j\,\omega)$) systems were first determined. Next, the uncertainty weight parameters from Equation (18) were estimated to satisfy Equation (17). Figure 10 provides a summary of the estimated parameters. The DC ($w_{\mathrm{DC},k}$) and high-frequency gains ($w_{\infty,k}$) of the filter were expressed in

percentage, representing the normalized system perturbation away from 1 on that frequency. Thus, 0 % of uncertainty indicates that the identified transfer function ($G_\ell$) from the blade effective wind speeds ($u_{\mathrm{beff}}$) to the corrected lidar-based inflow wind speeds ($u_{\mathrm{cor}}$) can have a gain of 1 in that frequency. Moreover, 10 % of uncertainty means that the identified transfer function ($G_\ell$) can have a gain of either 0.9 or 1.1 in that frequency.

### 3.3.1   Telescope parameters for no-induction case ($\mathrm{C_1}$)

The basic concept of the feedforward controller is the use of measured inflow wind speed from blade $i$ to control the blade and trailing edge flap angles at blade $i-1$. Assuming that the blades are rigid, constant rotor speed and pitch angle, and that Taylor's frozen turbulence hypothesis (Taylor (1938)) holds, resulted in a 1.7 s ($= \frac{2\pi}{3} \frac{30}{\pi\,\omega_\mathrm{r}}$, $\omega_\mathrm{r} = 11.74\,\mathrm{rpm}$) of preview time, or the time needed for blade $i-1$ to reach the position of blade $i$, i.e. 120° azimuth angle change. The simulation setup presented in Section 3.1 resulted in a hub-height mean wind speed of $13.06\,\mathrm{m\,s^{-1}}$. The assumption that the wind evolves

according to Taylor's frozen turbulence hypothesis, and with the induction zone effect being negligible, a focus distance of





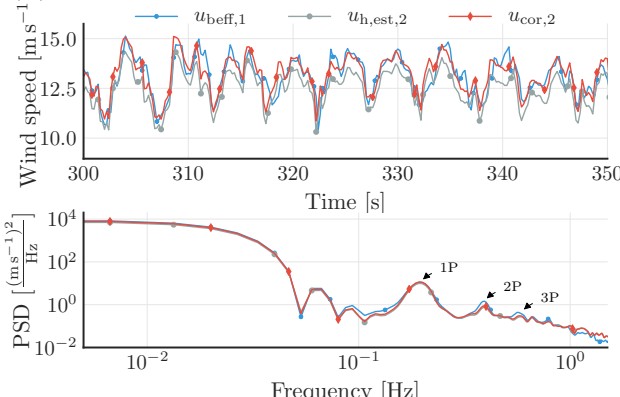

**Figure 11.** A sample time series of the blade effective wind speed from blade 1 ($u_{\text{beff},1}$) and the estimated ($u_{\text{h,est},2}$) and corrected ($u_{\text{cor},2}$) inflow wind speeds from blade 2 in the rotating frame of reference shown in the upper plot. The power spectral density (PSD) of the three signals is displayed in the lower plot.

22.2 m ($= 1.7\,\text{s} \cdot 13.06\,\text{m}\,\text{s}^{-1}$) was determined. In accordance with Bossanyi (2013) and Simley et al. (2014a), the inflow at 70 % ($\approx 44\,\text{m}$) of the blade radius could be assumed as most representative of the blade effective wind speed; hence, the telescope was located in this radial position. Specifically, the telescope orientation angles $\Phi_{\ell,i}$ and $\Gamma_{\ell,i}$ were found through the simulation, as the counter rotation of the blade segment angular orientation so that the lidar beam becomes parallel with the
rotor shaft axis (see Figure 2).

Figure 11 (upper plot) shows a sample time series of the blade effective wind speed from blade 1 ($u_{\text{beff},1}$), as well as the estimated ($u_{\text{h,est},2}$) and corrected ($u_{\text{cor},2}$) inflow wind speeds from blade 2. The three signals are in the rotating frame of reference. The lower plot displays the power spectral density (PSD) of the three signals. The dominant frequencies were clearly visible, as a result of the rotational sampling of the inflow wind speed by the blade-mounted telescope. The PSD analysis
highlighted these dominant frequencies as 1P, 2P, and 3P. Moreover, the plot revealed a good match at 1P between $u_{\text{beff},1}$ and $u_{\text{cor},2}$, although $u_{\text{cor},2}$ was slightly underestimated at higher harmonics.

We transformed the different blade effective and corrected inflow wind speeds from the rotating to the non-rotating frame of reference via the multiblade coordinate transformation ($T_{\text{mbc}}(\theta)$) according to Section 2.3. Afterward, we evaluated the PSD for the collective, yaw, and tilt components of the signals. Figure 12 displays the result. The plot highlights the absence of 1P
and 2P components (as observed in the rotating frame of reference, see Figure 11) in the non-rotating frame of reference, in line with Section 2.3. Below 0.1 Hz, a good match between the collective and tilt components could be observed, but with the yaw component of the corrected inflow wind speed ($u_{\text{cor,yaw}}$) slightly underestimated. Furthermore, the 3P component of $u_{\text{cor},k}$ (with $k \in \{\text{col, yaw, tilt}\}$) in the non-rotating frame of reference, which is the contribution of 2P and 4P from the rotating frame of reference, was likewise underestimated in all three components.





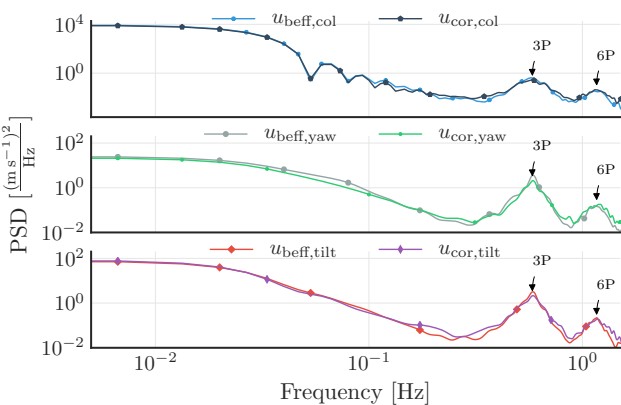

**Figure 12.** Power spectral density of the blade effective wind speeds ($u_{\mathrm{beff},k}$) and the corrected inflow wind speeds ($u_{\mathrm{cor},k}$) in the non-rotating frame of reference, with $k \in \{\mathrm{col}, \mathrm{yaw}, \mathrm{tilt}\}$.

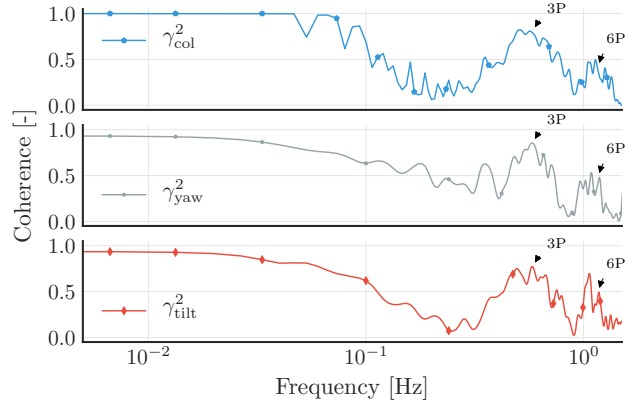

**Figure 13.** Coherences ($\gamma^2$) between the blade effective wind speeds ($u_{\mathrm{beff},k}$) and the corrected inflow wind speeds ($u_{\mathrm{cor},k}$) in the non-rotating frame of reference, with $k \in \{\mathrm{col}, \mathrm{yaw}, \mathrm{tilt}\}$.



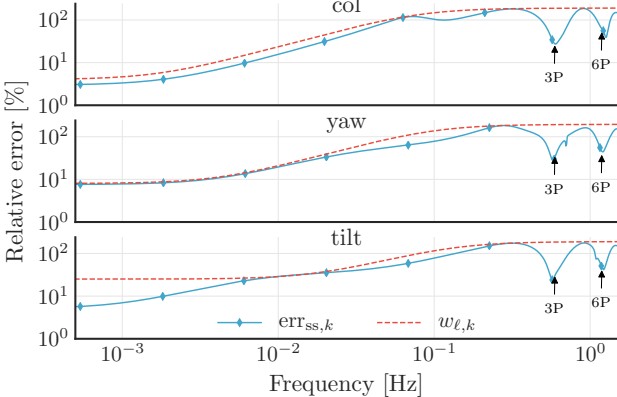

**Figure 14.** Relative error between the nominal plants ($G_{n,k}(j\,\omega)$) and those identified ($G_{\ell,k}(j\,\omega)$). The dashed line indicates the estimated frequency-dependent uncertainty weight ($w_{\ell,k}(j\,\omega)$), where $k \in \{\text{col, yaw, tilt}\}$.

Figure 13 reveals a good coherence on the frequencies where the powers were concentrated, i.e., below 0.1 Hz, and at 3P and 6P. Additionally, the plot discloses the declining coherence with frequency increase i.e., higher coherence was achieved at 0P than at 3P; the same could be implied between 3P and 6P. With Figure 11 highlighting the low-power content of the signals between 0P and 3P, and between 3P and 6P, low coherences were similarly seen at the same frequencies in Figure 13.

Furthermore, we determined the measurement uncertainty weights for the feedback–feedforward individual pitch control development and analysis. The blue lines in Figure 14 show the relative error between the resulting nominal ($G_{n,k}(j\,\omega)$) and identified ($G_{\ell,k}(j\,\omega)$) plants, in accordance with Equation (17). The uncertainty weight was approximated with a first-order minimum-phase filter (shown by dashed line), whose parameters from Equation (18) were labeled $C_1$ in Figure 10. Figure 14 shows a low uncertainty on the frequencies where the power of the signals were concentrated. Note that these uncertainties increased at higher harmonics.

### 3.3.2 Uncertainties around the no-induction telescope parameters ($C_2$)

In this section, we investigated the impact on the uncertainty weights when the telescope parameters cannot be selected as defined for the no-induction case, but somewhere close to these values. We carried out simulations involving a discrete set of sampled values for the focus distance, radial position of the telescope along the blade radii, and orientation angles of the telescope. The identified uncertainty weight parameters are labeled as $C_2$ in Figure 10. The plot shows that the crossover frequencies ($\omega_{0,k}$) for this case ($C_2$) either remain the same or decreasing slightly with respect to $C_1$. A significant increase was observed at the low-frequency (DC) uncertainties for the yaw and tilt components, i.e., the low-frequency uncertainties at the yaw and tilt components were changed from the no-induction case values of 8 % and 25 % to 20 % and 43 %, respectively. The high-frequency uncertainties remained nearly the same.



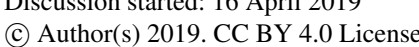


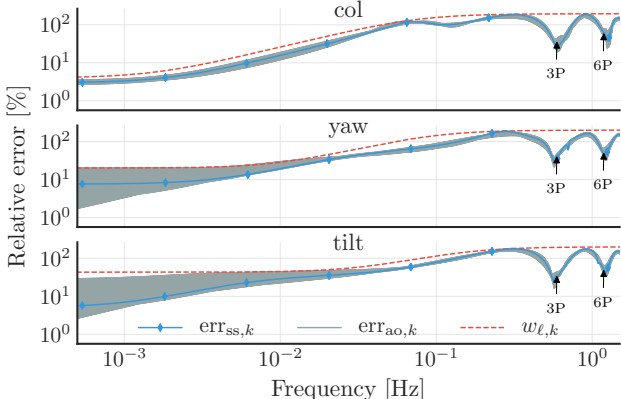

**Figure 15.** Relative errors between the nominal plants ($G_{n,k}(j\omega)$) and those identified ($G_{\ell,k}(j\omega)$) for a discrete set of sampled telescope parameters, where $k \in \{\text{col}, \text{yaw}, \text{tilt}\}$. The relative errors are represented with overlapping grey lines on the plot. The blue line with diamonds is the relative error found for the no-induction case ($\text{C}_1$).

In Figure 15, the overlapping grey lines represent the relative errors for the discrete set of sampled values. The blue line with diamonds represents the relative error found for the no-induction case ($\text{C}_1$). The plot underscored the occurrence of both a better and a worse set of telescope parameters that yield a lower or higher low-frequency uncertainty. For example, after performing a search, we found that the telescope parameters of $F = 20.2$ m, $R = 44.0$ m, $\Phi_{\ell,i} = -5.7°$, and $\Gamma_{\ell,i} = 9°$ would
5    result in the minimum value of $\sum_k \omega_{0,k}$, and the telescope parameters of $F = 28.2$ m, $R = 45.0$ m, $\Phi_{\ell,i} = -1.7°$, whereas $\Gamma_{\ell,i} = 5°$ would result in the maximum value of $\sum_k \omega_{0,k}$, where $k \in \{\text{col}, \text{yaw}, \text{tilt}\}$.

### 3.3.3 Optimal focus distance and available preview time ($\text{C}_3$)

To determine the optimal preview time, we kept the telescope parameters constant as defined in Section 3.3.1, except for the focus distance, which was allowed to vary between 20.2 and 30 m. Subsequently, we performed a search at assumed preview
10    times from 1.6 to 2.3 s, with a resolution of 0.2 s for each focus distance. Afterward, we evaluated the objective function from Equation (21) for all combinations of the focus distance and preview time. Figure 16 displays a plot, as a color map, of the result. Accordingly, the green line with the stars indicates the calculated preview time for the no-induction case, as determined by dividing the focus distance ($F$) with the hub-height mean wind speed ($\text{u}_{hh} = 13.06 \, \text{m s}^{-1}$). From Figure 14 in the no-induction case, the uncertainties ($w_{\ell,k}$) at the components were either close or above 100 % around 0.06 Hz; therefore, we
15    considered an $f_{\max}$ of 0.06 Hz in Equation (21). The optimal preview time could be determined for a given focus distance with the minimized objective function ($J$) in Equation (21). The blue line with the diamonds shows the resulting optimal preview time for a given focus distance.

Figure 16 shows these observations: (1) The values of the objective function increased with measurement distance from the rotor position. (2) The blue line with the diamonds emphasized that a higher preview time was available with respect to the no-




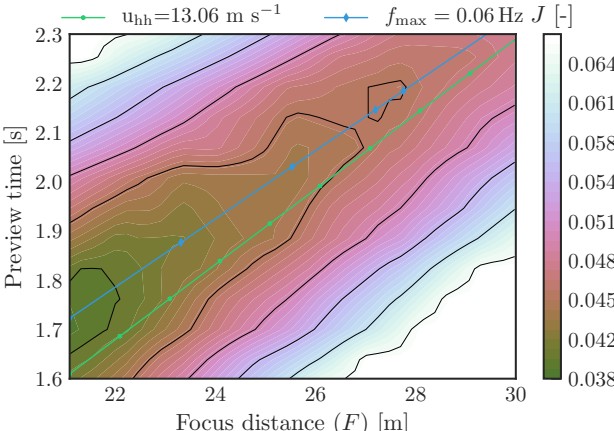

**Figure 16.** The optimal preview time for a given focus distance. The maximum frequency ($f_{\max}$) in the objective function ($J$) is set at 0.06 Hz. The green line with the stars is the calculated preview time for the no-induction case. The blue line with the diamonds is a linear fit of the optimal preview time determined for a given focus distance by considering $f_{\max}$ at 0.06 Hz in $J$.

induction case (green line with the stars), where the assumptions were (a) the blades are rigid, (b) Taylor's frozen turbulence hypothesis holds, and (c) induction effect is absent. (3) The preview time and focus distance were closely coupled; e.g., a changing focus distance implied a varying preview time.

To estimate the uncertainty weights for this case, we varied the focus distance of the lidar between 20.2 and 30 m with

1 m steps, while the other parameters were kept constant. A summary of the parameters of the uncertainty weights is given in Figure 10, denoted as $C_3$. By increasing the focus distance, the uncertainties at low-frequencies ($w_{DC,k}$) were increased to almost as much as at $C_2$ and were almost double those in the no-induction case ($C_1$).

As such, the results in this subsection highlight the following points:(1) A focus distance close to the rotor is more beneficial, and (2) the inflow wind slows down in front of the rotor due to the induction zone effect, which leads to a higher preview time

with respect to the no-induction case.

### 3.3.4 Telescope position along the blade span ($C_4$)

Bossanyi (2013) proposed that a blade-mounted lidar placed 70 % of the blade radius is most suitable for control input. We assessed in this subsection whether placing the blade-mounted lidar at 70 % ($\approx 44$ m) of the blade radius would result in the minimization of the objective function in Equation (21). We set $f_{\max}$ in Equation (21) as 0.06 Hz, while we maintained a focus

distance of 22.2 m. As such, the corresponding optimal preview time was 1.8 s, which is consistent with that of Section 3.3.3. Selection of the preview time plays an important role, as it affects the phase shift between the two signals in Equation (21). The corrected inflow wind speed measurement delayed with the preview time to align it with the blade effective wind speed.




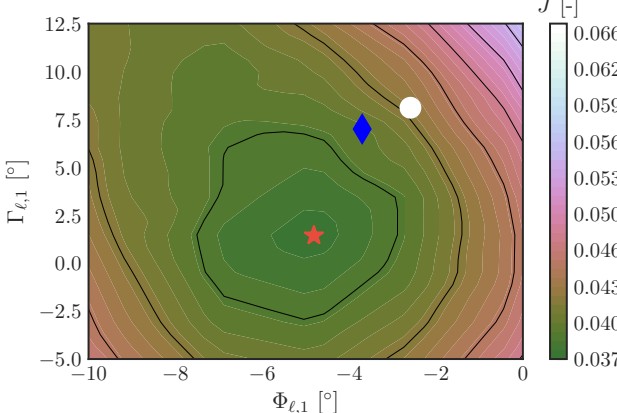

**Figure 17.** Optimal angular orientation of the telescope. The maximum frequency ($f_{\max}$) in the objective function ($J$) is set at 0.06 Hz. The orange star marks the optimum value; the blue diamond marks the initially chosen parameters; and the white dot marks the minimum point when the phase shift is neglected in Equation (21).

Indeed, with the telescope placed at 70 % of the blade span, the objective function in Equation (21) was minimized, confirming the findings of Bossanyi (2013). Moreover, varying the telescope radial position in a fairly small range (42–46 m) yielded a marginal increase (0.004) in the objective function value. A similar effect was observed on the identified uncertainty weight (marked as $C_4$ in Figure 10), which apparently obtained similar weights parameters as for the no-induction case ($C_1$), both of which below $C_2$.

### 3.3.5 Telescope orientation ($C_5$)

In this section, we evaluate whether the initially selected telescope orientation angles ($\Phi_{\ell,i}$ and $\Gamma_{\ell,i}$, with $i = 1, 2, 3$) would result in a minimized objective function in Equation (21). For this purpose, we fixed the telescope parameters as described in Section 3.3.1, with the exception of the orientation angles ($\Phi_{\ell,i}$ and $\Gamma_{\ell,i}$). The two angles were changed around the initially selected values. We simulated the lidar measurements with each new set of parameters. We used a preview time of 1.8 s for the post-processing, as discussed in Section 3.3.3, which was expected to result in a phase shift of approximately zero between the lidar measurement and the blade effective wind speed at low frequency, below the 1P frequency. We determined optimal orientation of the telescope in Figure 17 based on the objective function in Equation (21). In the plot, the blue diamond marks the initial telescope orientation based on the no-induction calculation, where $\Phi_{\ell,i} = -3.7°$ and $\Gamma_{\ell,i} = 7.0°$. The orange star indicates the obtained optimal value, where $\Phi_{\ell,i} = -4.8°$ and $\Gamma_{\ell,i} = 1.45°$. The discrepancy in $\Phi_{\ell,i}$ was quite small, however, with a higher degree of difference between the no-induction value of $\Gamma_{\ell,i}$ and the found optimal value of $\Gamma_{\ell,i}$. To understand such occurrence, we neglected the phase shift ($\varphi_k$) in Equation (21), as marked with a white dot, where $\Phi_{\ell,i} = -2.6°$ and $\Gamma_{\ell,i} = 8.0°$. Thus, the values became considerably closer to the values based on the no-induction calculations.



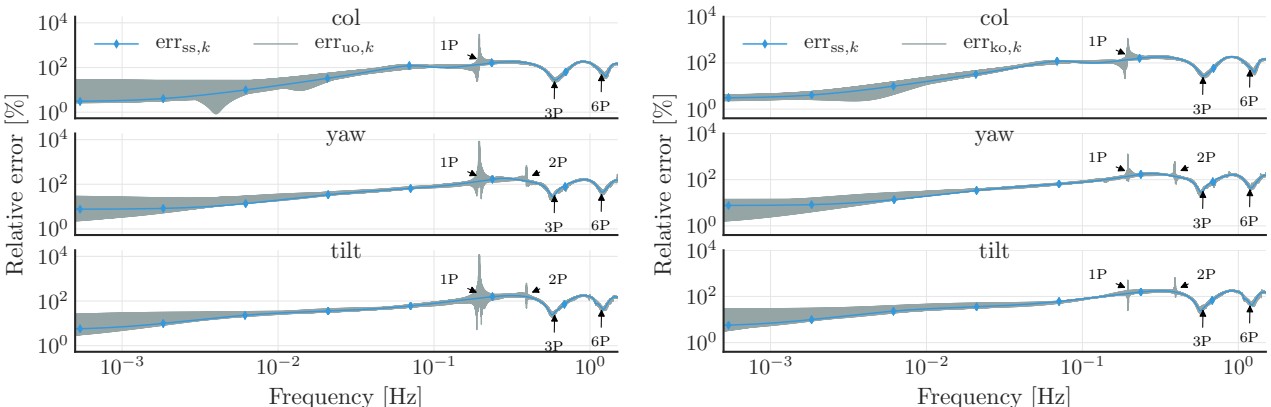

(a) Unknown telescope orientation misalignment.    (b) Known telescope orientation misalignment.

**Figure 18.** Relative errors between the nominal plants ($G_{\mathrm{n},k}(j\omega)$) and those identified ($G_{\ell,k}(j\omega)$) for the discrete set of sampled telescope parameters with unknown and known telescope orientation misalignment, where $k \in \{\text{col, yaw, tilt}\}$. The relative errors are represented with overlapping grey lines on the plot. The blue line with diamonds is the relative error found for the no-induction case ($C_1$).

The identified frequency-dependent uncertainty weights parameters were labeled as $C_5$ in Figure 10. After the orientation angles were changed by $\pm 3°$ around the no-induction values, the identified uncertainty weight parameters for this case were still close to the values found for the no-induction case.

### 3.3.6 Telescope orientation misalignment ($C_6$)

In this subsection, the uncertainty weight parameters were identified for the cases where a single or two of the telescopes have been aligned differently, whereas their values corresponding to the no-induction case were obtained. Such cases could occur, for example, during telescope installation. Initially, we assumed this misalignment as unknown but detectable to allow for the accurate correction of the lidar-based inflow wind speed measurement. To simulate these cases, we fixed the telescope parameters as described in Section 3.3.1, except for the orientation angles of $\Phi_{\ell,i}$ and $\Gamma_{\ell,i}$ of the telescopes mounted on

the second and third blades. The angular values were changed around the no-induction values by $\pm 5°$ ($\Phi_{\ell,i} = \Phi_{\ell,1} \pm 5°$ and $\Gamma_{\ell,i} = \Gamma_{\ell,1} \pm 5°$ with $i = 2,3$) as follows. First, the value was changed only for the telescope mounted on the second blade, then for the telescopes mounted on the second and third blades.

 We evaluated such setups in the simulations. Figure 18 displays the relative errors between the nominal and the identified systems. Figure 18a reveals a 1P peak at the collective component and 1P and 2P peaks at the yaw and tilt components. As

shown in Section 3.2, adding a phase shift of $1°$ to the 1P harmonic and reducing the DC offset for one of the signals in the rotating frame of reference would result in such undesired higher harmonic peaks at the collective, yaw, and tilt components in the non-rotating frame of reference. Figure 18b underlines that, by assuming that the misalignment angles is identifiable and that the lidar-based inflow wind speed measurement is corrected accordingly, the undesired peak at 1P was reduced almost with



one decade, although existent on all the components. Furthermore, the low-frequency uncertainties were reduced significantly on all three components.

## 4   Discussion

We showed that the determined telescope parameters with assumptions of rigid blades, absence of induction, and Taylor's
frozen turbulence hypothesis hold, provide a good trade-off between simplicity and accuracy. However, we would like to emphasize the presence of uncertainties in all three components, as the result of the wind evolution, the simplicity of the induction zone correction, "cyclops dilemma", and using only a single-point measurement for the estimation of the blade effective wind speed at assumed zero value of $v_{h,i}$ and $w_{h,i}$ components (see Section 2.1), etc. Therefore, it is important to consider the uncertainties in the controller development; e.g., uncertainty at the yaw and tilt components was already approximately 150 % at
0.195 Hz (1P frequency), which could have affected the performance of the controller, i.e., it can lead to increased values of the sensitivity function, causing load increase on the non-rotating components of the wind turbine, as was asserted by Ungurán et al. (2019).

The results show that the measurement uncertainties increase with distance from the rotor plane. Therefore, a closer measurement of the inflow wind speed to the rotor plane is preferred. Note that control development must proceed with sufficient
attention so as to ensure that the feedforward controller does not result in higher time delay than the available preview time. For example, a feedforward controller with a crossover frequency of 0.06 Hz may result in higher time delay compared to that with a crossover frequency of 0.1 Hz (Dunne and Pao (2016)). With this, we want to point out that the feedforward controller crossover frequency and the focus distance are coupled. Hence, defining the former typically leads to a minimal selectable focus distance.

As stated above, the lidar and telescope parameters based on the assumptions we made in Section 3.3.1 provide a good trade-off between simplicity and accuracy. They are close to the optimal parameters we found for the discrete set of sampled values of the focus distance, the radial position of the telescope along the blade, and the orientation angles of the telescope, as shown in Section 3.3.2. Nevertheless, this is not the case for the preview time; the rotor blocking effect increases the available preview time from 1.7 s to 1.8 s. The crossover frequency of the feedforward controller affects the time delay. With a higher preview
time available, we can select a lower crossover frequency, e.g., where uncertainty is still below 100 %, for the feedforward controller. Note that such uncertainty is defined as the normalized system perturbation away from 1 on that frequency; hence, it can be higher than 100 %. This understanding gives us more room during the feedback–feedforward control development. We established a method for estimating the available preview time, which can be extended in field tests for that purpose, as well as to delay the feedforward control signal accordingly. This can be done by an online evaluation of Equation (21), for example,
using the last ten minutes estimated blade effective wind speeds and the corrected inflow wind speeds, and then carrying out a similar search we proposed in Section 3.3.3.

We found that the blade-mounted lidar placed at the 70 % of the blade radius results in a minimum of the objective function in Equation (21). This finding is consistent with the conclusion of Bossanyi (2013) for a blade mounted lidar and is in line with



the findings of Simley et al. (2014a) for the hub-mounted lidar system. The phase shift in the objective function in Equation (21) acts as a fine tuning of the available preview time. We aligned the two signals with the assumption that the measured inflow wind speed hits the wind turbine after 1.8 s, as we found in Section 3.3.3. The signals were sampled with a sampling time of 0.2 s, which limits the fine tuning of the available preview time. Note that LES simulations with lower sampling time are
resource and time expensive. When we neglected the phase shift from the objective function, the obtained orientation angles were considerably closer to the orientation angles, based on the no-induction case. Such a small deviation was expected with respect to the assumptions we made during the calculation of the values for the no-induction case (see Section 3.3.1).

Any unknown orientation angle misalignment for one of the telescopes leads to an unknown contribution of the rotational speed to the lidar-based line-of-sight wind speed measurement. This is the reason for the increase in the uncertainty in the
collective component from 3 % to 27 %. Nevertheless, this can be reduced to 4 % by assuming that we are able to detect the angular offset. In addition, an undetected misalignment of the telescope orientation angle results in a phase shift of the 1P harmonic and a reduction or increase of the DC offset of the signal in the rotating frame of reference. This subsequently leads to undesired peaks at 1P and 2P frequencies at the collective, yaw, and tilt components in the non-rotating frame of reference, or to nearly 10,000 % of high-frequency uncertainties, which we were able to reduce to 1,000 %, but not completely eliminate.
Thus, the question as to whether robust stability and performance can be ensured with such a high uncertainty still remains, i.e., Ungurán et al. (2019) assumed only 300 % of high-frequency uncertainties on the yaw and tilt components. To avoid such a peak, the telescopes need to be well aligned with each other and the blade segment orientation angles and linear velocities should be measured well. We showed in Sections 3.2 and 3.3.6 that an unknown orientation angle misalignment leads to a 1P peak at the yaw and tilt components in the frequency domain. Therefore, the orientation angles of the telescope can be identified
by formulating an optimization problem, whose main objective is to minimize the 1P peaks at the yaw and tilt components with orientation angles of the telescopes as the decision variables.

As we considered only one reference wind turbine with a single inflow wind condition, we need to assess how the uncertainties change for other wind turbines with different wind speeds, turbulence intensities, yaw misalignments, etc. The uncertainty weight found in Section 3.3.2 might cover these cases, and may serve helpful in the control development rather
than that found in the no-induction case in Section 3.3.1. Nevertheless, this may further reduce the gains of the feedforward controller (see Ungurán et al. (2019)) with respect to the controller developed by using the uncertainty weight found for the no-induction case, thus limiting the benefits of the lidar system.

We modeled the uncertainty weight as first-order minimum-phase filters. On this regard, if robust performance and stability is not ensured by the use of this weight, then a higher order filter could be studied to observe the relative error over the frequency
more closely, e.g., for the tilt component in Figure 14.

The uncertainty weight identified for the no-induction case can be directly included into robust feedback–feedforward individual pitch and trailing edge flap control development to guarantee robust stability and performance. If for some reason one or more lidar and telescope parameters cannot be selected as for the no-induction case, but close to these values, the uncertainty weight from $C_2$ can be used for robust feedback–feedforward control development. However, this might lead to a conservative





feedforward controller with respect to performance, i.e., the low-frequency gains of the feedforward controller will be reduced, as highlighted by Ungurán et al. (2019).

The methodology we presented in this paper can be applied in identifying the uncertainty weight for higher harmonics control development, i.e., selecting $n_{\mathrm{h}}$ as 2 in Equation (8) can be used to identify the uncertainty weight for the controller
developed to mitigate the 2P dynamic blade loads.

## 5   Conclusion

Our paper aimed to identify the blade-mounted lidar measurement uncertainties as frequency-dependent uncertain weights that can be employed in feedback–feedforward individual pitch and trailing edge flap control development and analysis.

Typically, induction zone increases the preview time, thus, the latter must be taken into account in the control development
and implementation. We presented a method that can estimate the preview time online; hence, the control signal can be delayed accordingly. We found that an inflow wind speed measurement close to the rotor plane is preferable, which emphasizes the influence of the wind evolution, the further the measure takes place from the rotor plane the more the wind develops until it reaches the rotor. However, the selected focus distance should provide sufficient preview time so that the time delay introduced by the feedforward controller and actuators could be eliminated. This sets the lower limit for the selectable focus distance.
Accordingly, we introduced a simple method to calculate the telescope and lidar parameters. Nevertheless, we showed in a large-eddy simulation, that such approach provides a good trade-off between a fast-forward determination of the telescope parameters and accurate inflow wind speed measurement. Measurement uncertainties were present due to wind evolution, "cyclops dilemma", using single-point measurement to estimate the blade effective wind speed, and the assumptions we made to correct the measurements. The uncertainty weight, as we have identified in this paper, can be directly included in the robust
feedback–feedforward individual pitch and trailing edge flap control development to ensure robust stability and performance. However, to prevent the transfer functions ($G_\ell$) from the blade effective wind speeds ($u_{\mathrm{beff}}$) to the corrected lidar-based inflow wind speeds ($u_{\mathrm{cor}}$) from having a large high-frequency gain of more than 11, which would result in more than 1,000 % high-frequency uncertainties, the telescopes must be well aligned with each other and the blade segment orientation angles and linear velocities should be measured well.

*Acknowledgements.* This work has been partly funded by the Federal Ministry for Economic Affairs and Energy according to a resolution by the German Federal Parliament (projects DFWind 0325936C and SmartBlades2 0324032D). The support of a fellowship from Hanse–Wissenschaftskolleg in Delmenhorst (HWK) is likewise gratefully acknowledged.




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
