# Peer review of "Uncertainties identification of the blade-mounted lidar-based inflow wind speed measurements for robust feedback-feedforward control synthesis"

_Wind Energy Science, 2019_

## Referee Comment (RC1) · Anonymous Referee #1 · 15 May 2019

The paper is well written and provides a good synthesis for understanding the uncertainties that come up when using blade mounted lidars.

On page 5, line 5 it is stated that "the rotational effect of the blade was not accounted for ..." I am just wondering if you can add a brief explanation as to why it isn't accounted for. Also, what about yaw of the wind turbine? I assume this was all done without considering what would happen if the turbine yaws to a new wind direction. This is probably something that can be ignored, but it was something that got me thinking as an interesting problem to try to tackle although outside the scope of this paper.

[Figure]

On page 8, line 5 it says in this sentence that blade root flapwise and edgewise moments are widely available wind turbine sensors, however in my experience these sensors are found on most research turbines, but not on utility wind turbines in the industry.
* * *

---

## Referee Comment (RC2) · Anonymous Referee #2 · 25 May 2019

The authors present an uncertainties identification study for blade-mounted lidar wind speed measurements. In general, this is an interesting topic since using lidar for control is a promising technology. Based on the abstract and the conclusions, there are 3 main contributions:

1. Identification and modeling of uncertainties as frequency-dependent uncertain weights that can be employed in feedback–feedforward individual pitch and trailing edge flap control development and analysis.

2. Presentation of a method that can estimate the preview time online

3. Introduction of a simple method to calculate the telescope and lidar parameters.

The paper simulates three blade-mounted lidar systems. Most of the paper focuses on the analysis between two set of signals, where $k \in \{$col,yaw,tilt$\}$:

1. $u_{\text{cor},k}$: processed signals from the blade-mounted lidar systems.

2. $u_{\text{beff},k}$: blade-effective wind speeds transformed into fixed frame.

The paper is mostly well written and the amount of work done is very impressive. Performing several large-eddy simulations created a nice environment for the intended contributions.

However, there are some issues in the analysis:

**Uncertainties calculation**

This is the main weakness of the work in my opinion. The issue and can be separated in two sub-problems:

1. The modeling and identification of the transfer function is not consistent.

2. The transfer function is used for a measure of uncertainty, which is not correct.

Inconsistency

Based on Figure 5 and Equation (11), the transfer function between $d$ and $d_\Delta$ is $I + \Delta_\ell W_\ell$. $\Delta_\ell$ seems to be a multiplicative uncertainty used in robust control to transform a system into a M-$\Delta$ structure. Usually this structure is then used to design

controllers which are still stable for a multiplicative uncertainty $\Delta_\ell \leq 1$. For this purpose, the uncertainty weight needs to be identified using Equation (17) as a worst case. Here, the nominal model should be used, namely $I$. However, the authors use a first order low-pass filter, without further explications. Anyway, the whole identification of the uncertainty weight can be considered as a fit to $1 - G_\ell$. Please note, the uncertainty weight is not the multiplicative uncertainty $\Delta_\ell$.

Further, in the caption of Figure 5, $G_{\mathrm{d,f}}$ is named "disturbance model" and $G_{\mathrm{wt,f}}$ is named "wind turbine model". However, both should be part of the wind turbine: $G_{\mathrm{d,f}}$ is the part of the wind turbine which models how the disturbance affects the outputs. $G_{\mathrm{wt,f}}$ is the part of the wind turbine which models how the control inputs affect the outputs.

Measure of uncertainty

The author write on page 16: "Thus, 0% of uncertainty indicates that the identified transfer function ($G_\ell$) from the blade effective wind speeds ($u_{\mathrm{beff}}$) to the corrected lidar-based inflow wind speeds ($u_{\mathrm{corr}}$) can have a gain of 1 in that frequency. Moreover, 10% of uncertainty means that the identified transfer function ($G_\ell$) can have a gain of either 0.9 or 1.1 in that frequency."

In my opinion, the use of uncertainty is misleading here. If the uncertainty weight is 0 at a certain frequency, this means that the gain of the identified transfer function is equal to the nominal transfer function. If the uncertainty weight is 10% at a certain frequency, this means that the gain of the identified transfer function is within 10% of the gain of to the nominal transfer function. However, it does not give you any information about an uncertainty in the sense how well a lidar measures or not or how well the signal can be used for feedforward control.

Small example: Let's considered two signals, $s_1$ and $s_2$, where $s_2$ is generated by

passing signal $s_1$ through a linear low-pass filter. Applying the approach above will lead to a uncertainty weight of 0 at 0 Hz and will approach 1 for high frequencies. If now $s_2$ is the disturbance $d$ acting on the plant and $s_1$ the signal $d_\Delta$ used for feedforward control, one could simply use the same linear low-pass filter to get perfect disturbance rejection.

In short: I fear the proposed method is not useful to describe the uncertainty for lidar measurements.

Please check the uncertainty modeling in your reference Dunne and Pao 2016 (using additional noise input) or the measurement error introduced in E. Simley and L. Pao, "Reducing LIDAR wind speed measurement error with optimal filtering," 2013 American Control Conference, Washington, DC, 2013, pp. 621-627..

**Preview time estimation**

Section 2.6 describes the procedure how the preview time is estimated. Here, the phase angle between $u_{\mathrm{cor},k}$ and $u_{\mathrm{beff},k}$ is used. It is not well explained, but still understandable that minimizing the absolute phase angle provides signals which are well aligned in time. Further, the weighting with the spectra $S_k$ is a quite empirical approach, but might be considered to be an acceptable approach to estimate the preview time. However, dividing with the coherence seems strange to me. Since the coherence can become zero, this does not seem right. In my opinion, it also does not help much that later you explain that only frequencies up to 0.06 Hz are used, where the coherence is larger than zero. The use of the coherence in $J$ is not explained. It also is not included in the integral in the denominator, so also can not be considered an empirical weight. It seems to be an additional, not well explained and maybe not necessary complexity. It is not clear why not usual methods to determine the preview time are used, such as the peak of the cross-correlation. In Held, D. P. and Mann, J.: Lidar Estimation of Rotor-

Effective Wind Speed – An Experimental Comparison, Wind Energ. Sci. Discuss. in review, 2019. the information theoretical delay estimator presented in Moddemeijer(1988) is proposed, which also seems to be more useful. Further, it is not clear how this method can "estimate the preview time online" as claimed in second of the three main contributions of the paper.

Further, $J$ is used in Figure 17 and Section 3.3.5 to optimize the telescope orientation. Lidar scan configuration has been done in several studies before based on different cost functions. Minimizing $J$ with a fixed preview time might lead to somehow optimal telescope orientation angles for the selected preview time in terms of timing. However, it is not clear, how the optimization leads to useful signals with high measurement quality if e.g. the mean wind speed is chancing etc. Further, the method (running LES simulations and using $J$) does not seem to be a "simple method to calculate the telescope and lidar parameters" as claimed in the third of the three main contributions of the paper.

**Organization**

The paper's organization can be improved by following points:

- Section 2.1 and 2.2. In these two sections, the lidar-simulation, the estimation of the blade-effective wind speed and the definition of the blade-effective wind speed are somehow mixed together. This was quite confusing to me. It is very important to understand, how the two sets of signals mentioned above have been obtained, since the whole study focuses on the analysis between them. It would be better to have three subsections:

    - New Section 2.1: lidar-simulation: all the text of Section 2.1 until page 6, line 9.

- – New Section 2.2: wind speed estimation: all the rest of Section 2.1 and text of Section 2.2 on page 7.
- – New Section 2.3: blade-effective wind speed: text of Section 2.2 on page 6.

- Similarly, in Section 2.3, you could also explain that MBC is also applied to the blade-effective wind speed.

- The paragraph about the control development (page 9), the remarks, the $G_{\mathrm{d,f}}^{-1}$ and the performance weight is not important for the rest of the paper and should be removed. Again, it seems to be an additional, not well explained and unnecessary complexity.

- Equation (6) and (7): Since the whole paper focus on the two sets of signals, Function $f$ should be either explained in detail or simply avoided. Again, it seems to be an additional, not well explained and maybe not necessary complexity.

- Section 3.1 explains the simulation setup using PALM, which then seems to be used in Section 3.3. In Section 3.2 however, generic wind speed measurements are used. It would be better in my opinion to switch them.

**Minor issues**

- Page 6, line 12: to estimate $u_{\mathrm{h,est},i}$ from Equation (1) to (3), you also need to neglect the weighing function. This is missing in the assumptions leading to Equation (4). Further, the expression "the measured LOS can be corrected" might be misleading, since the LOS are correct, you use Equation (4) to estimate or reconstruct the longitudinal wind speed.

- Several variables are introduced relatively late, e.g. $k$, $V_i(\xi)$.

- The variables are not consistently named: you use "blade-effective wind speed" for (1) the original $u_{\mathrm{beff},i}$ with $i$ for blade 1, 2, and 3, as well as (2) for the transformed $u_{\mathrm{beff},k}$, for $k \in \{$col,yaw,tilt$\}$.

- Section 2.6: It is not clear that $u_{\mathrm{cor},k}$ is delayed. The only delay introduced in Section 2.3 is for the pitch angles.

- The simulation time is not stated in Section 3.1, but might be interesting for all the frequency estimates. Sorry, if I missed that information somewhere else.

- Page 12, line 12: $and$ not necessary.

---

## Author Comment (AC1) · 14 Jul 2019

We thank the referee for the attention and enormous energy to read our work and write a detailed review which has helped us to improve the content of the paper. We attached a zip file which includes our response to reviewer comments. Furthermore, an additional document is also attached, highlighting the changes that have been made.

Please also note the supplement to this comment:

[Figure]

https://www.wind-energ-sci-discuss.net/wes-2019-15/wes-2019-15-AC1-supplement.zip

---

## Author Response (AR1)

**Response to the first reviewer comments (RC1)**

Róbert Ungurán, Vlaho Petrović, Lucy Y. Pao, Martin Kühn

We thank the referee for the attention and enormous energy to read our work and write a detailed review which has helped us to improve the content of the paper. This document includes our response to reviewer comments. Furthermore, an additional document is attached, highlighting the changes that have been made.
* * *
*Q 1: On page 5, line 5 it is stated that "the rotational effect of the blade was not accounted for ..." I am just wondering if you*
5    *can add a brief explanation as to why it isn't accounted for. Also, what about yaw of the wind turbine? I assume this was all done without considering what would happen if the turbine yaws to a new wind direction. This is probably something that can be ignored, but it was something that got me thinking as an interesting problem to try to tackle although outside the scope of this paper.*

**Reply**: This is a very interesting question. Yawing the wind turbine would be seen in the blade segment velocities, so in the
10    wind speed correction, it is accounted for. However, if there is yaw error, there is effectively more crosswise wind velocity, so the way we estimated the longitudinal velocity would have larger error. Yaw error could also be estimated to try to correct for this, but this was out of the scope of this paper.

We assumed that we have an instantaneous single point measurement, because its effect is considered neglegible, e.g. ZephIR lidar can have a sampling rate up to 400 Hz, so the accumulation time is smaller than 2.5 milliseconds. This would lead to a
15    swept distance of 17.2 cm ( 11.74 rpm $\cdot \frac{2\pi}{60} \cdot 0.0025\,\text{s} \cdot 80\,\text{m} \cdot 70\%$ span). We assumed that the wind speed for this small swept distance is approximately constant.
* * *
*Q 2: On page 8, line 5 it says in this sentence that blade root flapwise and edgewise mo- ments are widely available wind turbine sensors, however in my experience these sensors are found on most research turbines, but not on utility wind turbines in the industry.*

20    **Reply**: To our best knowledge, some wind turbine manufacture includes such a sensor in the series production, while others offer them on demand. We replaced the wording in this sentence as follows:

*To account for this effect in the lidar-based inflow wind speed measurement, we construct a second-order polynomial function*
*($f$), whose inputs are chosen as rotor speed ($\omega_r$), blade pitch angle ($\beta_i$), and blade root flapwise and edgewise moments*
*($M_{fw,i}$, $M_{ew,i}$). Rotor speed and blade pitch angles are easily measured, and we assumed that the blade root flapwise and*
25    *edgewise moment sensors are also available for implementing this method.*

**Response to the second reviewer comments (RC2)**

Róbert Ungurán, Vlaho Petrović, Lucy Y. Pao, Martin Kühn

We thank the referee for the attention and enormous energy to read our work and write a detailed review which has helped us to improve the content of the paper. This document includes our response to reviewer comments. Furthermore, an additional document is attached, highlighting the changes that have been made.

**Uncertainties calculation**

**Q 1:** *The nominal model should be used, namely I. However, the authors use a first order low-pass filter, without further explications.*

**Reply**: The gain of the low-pass filter over the frequency of interest was 1. Using I for the low-pass filter leads to the same result. Due to also the second point "The transfer function is used for a measure of uncertainty, which is not correct" we have updated our modelling and have added some text to clarify this (see Section 2.6).

**Q 2:** *Please note, the uncertainty weight is not the multiplicative uncertainty $\Delta_\ell$. The transfer function is used for a measure of uncertainty, which is not correct.*

**Reply**:

[Figure]

**Figure A.** Block diagram of the disturbance rejection control design with performance weight and uncertain disturbance measurement. $K_{\mathrm{fb,f}}$, $K_{\mathrm{ff,f}}$ are the feedback and feedforward controllers, $G_{\mathrm{wt,f}}$ is the wind turbine model from the control input to output, $G_{\mathrm{d,f}}$ is the wind turbine model from the disturbance to the output, $G_{\mathrm{n,f}}$ is the nominal disturbance measurement model, $\Delta_\ell$ is the uncertainty, $W_\ell$ is the measurement uncertainty weight, and $W_{\mathrm{p}}$ is the performance weight. The f in the index refers to the non-rotating (fixed) frame of reference.

This is a very good point and we have changed the modelling in accordance with Figure A, where we added a nominal disturbance measurement transfer function ( $G_{\mathrm{n,f}}$) in parallel to the uncertainty weight ($W_\ell$), leading to an additive disturbance measurement uncertainty modelling.

[Figure]

(a) $C_1$: ideal modeling accuracy no-induction case.

(b) $C_2$: more realistic modeling accuracy, with uncertainties around the no-induction case

**Figure B.** The identified disturbance measurement transfer functions ($G_{\ell,k}(j\omega)$). The dashed-dotted line indicates the estimated nominal disturbance measurement models ($G_{n,k}(j\omega)$). The dashed line shows the sum of the estimated nominal disturbance measurement models and uncertainty weights ($G_{n,k}(j\omega) + w_{\ell,k}(j\omega)$), where $k \in \{\text{col, yaw, tilt}\}$.

We repeated our investigation and first we established the transfer functions ($G_\ell$) from the blade effective wind speeds ($u_{\text{beff}}$) to the corrected lidar based inflow wind speeds ($u_{\text{cor}}$), then we separately identified the nominal disturbance measurement model ($G_{n,k}(j\omega)$) and the uncertainty weight ($w_{\ell,k}(j\omega)$) as a 5th-order minimum phase filter for each of the inputs in such a way as to satisfy the following inequalities

$$|G_{n,k}(j\omega)| \quad < \quad |G_{\ell,k}(j\omega)|, \, \forall \omega, \tag{1}$$

and

$$|G_{n,k}(j\omega) + w_{\ell,k}(j\omega)| \quad > \quad |G_{\ell,k}(j\omega)|, \, \forall \omega, \tag{2}$$

with $k \in \{\text{col, yaw, tilt}\}$.

For example, this led to the results shown in Figure B. The figure highlights that $G_{n,k}$ and $G_{n,k}+w_{\ell,k}$ are the lower and upper bounds of $G_{\ell,k}$. Although $C_1$ is useful to see how good our disturbance measurement (how far the magnitude of $G_{\ell,k}(j\omega)$ is from 1), it is too optimistic, any variation in the telescope parameters or inflow wind condition could result in $G_{\ell,k}(j\omega)$ being outside of the bounds. In contrast, $C_2$ covers a wide range of telescope parameter variation, and hence, if for some reason one or more lidar or telescope parameters cannot be selected as for the no-induction case, but close to these values, the established transfer functions from $C_2$ can be used for robust feedback–feedforward control development. In addition, $C_2$ also covers the cases where the mean blade pitch angle is increased or decreased because the wind turbine is at a different operating point.

The updated modelling is described in Section 2.6 and it also addresses the "Measure of uncertainty" comments of the referee.
* * *
**Q3:** *Further, in the caption of Figure 5, $G_{d,f}$ is named "disturbance model" and $G_{wt,f}$ is named "wind turbine model". However, both should be part of the wind turbine: $G_{d,f}$ is the part of the wind turbine which models how the disturbance affects the outputs. $G_{wt,f}$ is the part of the wind turbine which models how the control inputs affect the outputs.*

**Reply**: We named these based on Skogestad and Postlethwaite (2005), where they call the "disturbance model" as the transfer function from the disturbance to the output and the "plant model" as the transfer function from the control signal to the output. But our naming could be confusing, hence, we change the caption of Figure 5 as:

*Block diagram of the disturbance rejection control design with performance weight and uncertain disturbance measurement. $K_{fb,f}$, $K_{ff,f}$ are the feedback and feedforward controllers, $G_{wt,f}$ is the wind turbine model from the control input to output, $G_{d,f}$ is the wind turbine model from the disturbance to the output, $G_{n,f}$ is the nominal disturbance measurement model, $\Delta_\ell$ is the uncertainty, $W_\ell$ is the measurement uncertainty weight, and $W_p$ is the performance weight. The f in the index refers to the non-rotating (fixed) frame of reference.*

**Preview time estimation**
* * *
**Q4:** *Section 2.6 describes the procedure how the preview time is estimated. Here, the phase angle between $u_{cor,k}$ and $u_{bef,k}$ is used. It is not well explained, but still understandable that minimizing the absolute phase angle provides signals which are well aligned in time. Further, the weighting with the spectra $S_k$ is a quite empirical approach, but might be considered to be an acceptable approach to estimate the preview time. However, dividing with the coherence seems strange to me. Since the coherence can become zero, this does not seem right. In my opinion, it also does not help much that later you explain that only frequencies up to 0.06 Hz are used, where the coherence is larger than zero. The use of the coherence in $J$ is not explained. It also is not included in the integral in the denominator, so also can not be considered an empirical weight. It seems to be an additional, not well explained and maybe not necessary complexity. It is not clear why not usual methods to determine the preview time are used, such as the peak of the cross-correlation.*

**Reply**: Thank you for this comment, we had added the coherence as a weight, but we missed including it in the denominator. We wanted to give more emphasis to the phase shift where the coherence is high. We agreed that this was an unnecessary complexity. We have switched to the method you suggested (cross-correlation) which is a more straightforward way to determine the preview time.

The updated Section 2.7 is as follows:

*Preview time plays an important role in the development of feedforward control. It must be larger than or equal to the time delay introduced by the feedforward controller and actuator dynamics. It is preferable to be equal, but a larger value is acceptable, as additional time delay can be easily introduced into the feedforward controller, as shown in Figure 4. To determine the optimal preview time for a given focus distance, we evaluated the cross-coorelation between the blade effective*

($u_{beff,k}$) and the corrected inflow ($u_{cor,k}$) wind speeds, with $k \in$ {col, yaw, tilt}, and we chose the index of the peak value as the available preview time.

Furthermore, we updated the Discussion section, with the following:

*By evaluating the cross-coorelation between the blade effective ($u_{beff,k}$) and the corrected inflow ($u_{cor,k}$) wind speeds for a discrete set of sampled values of the focus distances in Section 3.3.3, we found that the preview time is constant for all the selected focus distances. It is closely coupled to the time needed for blade $i-1$ to reach the position of blade $i$, i.e. 120° azimuth angle change. For example, by considering laminar inflow with wind shear, no matter what the focus distance is, the delay time between the corrected inflow wind speeds from blade 1 and the blade effective wind speed from blade 3, will always be the same, which is the time needed for blade $i-1$ to reach the position of blade $i$. If the focus distance has changed, the $\phi$ in the MBC transformation also has to be changed, furthermore, the control signal should be delayed accordingly. Note that control development must proceed with sufficient attention so as to ensure that the feedforward controller does not result in higher time delay than the available preview time. For example, a feedforward controller with a crossover frequency of 0.1 Hz may result in higher time delay compared to that with a crossover frequency of 0.2 Hz (Dunne and Pao (2016)). With this, we want to point out that the feedforward controller crossover frequency and the focus distance are coupled. Hence, defining the former typically leads to a minimal selectable focus distance.*
* * *
**Q 5:** *Further, J is used in Figure 17 and Section 3.3.5 to optimize the telescope orientation. Lidar scan configuration has been done in several studies before based on different cost functions. Minimizing J with a fixed preview time might lead to somehow optimal telescope orientation angles for the selected preview time in terms of timing. However, it is not clear, how the optimization leads to useful signals with high measurement quality if e.g. the mean wind speed is chancing etc.*

**Reply**: In the revised manuscript, to analyse what the optimal telescope parameters should be for a given focus distance, we introduce a new objective function in Section 2.8, where the objective function is based on the coherence ($\gamma_k^2$) between the blade effective ($u_{\mathrm{beff},k}$) and the corrected inflow ($u_{\mathrm{cor},k}$) wind speeds, with $k \in$ {col, yaw, tilt}, leading to the following objective function

$$J_{\mathrm{lp}} \;=\; \sum_k J_{\mathrm{lp},k} \;=\; \sum_k \gamma_k^2(f) \; . \tag{3}$$

By evaluating $J_{\mathrm{lp}}$ for a discrete set of sampled lidar and telescope parameters, the maximum of the objective function would result in the optimal telescope parameters within the discrete set of sampled lidar and telescope parameters.
* * *
**Q 6:** *Further, the method (running LES simulations and using J) does not seem to be a "simple method to calculate the telescope and lidar parameters" as claimed in the third of the three main contributions of the paper.*

**Reply**: The telescope parameters computation is based on a simple method described in Section 3.3.1. We have used LES to validate the approach and model the nominal transfer function and uncertainty weight.

**Organization**
* * *
*Q 7:* *Section 2.1 and 2.2. In these two sections, the lidar-simulation, the estimation of the blade-effective wind speed and the definition of the blade-effective wind speed are somehow mixed together. This was quite confusing to me. It is very important to understand, how the two sets of signals mentioned above have been obtained, since the whole study focuses on the analysis between them. It would be better to have three subsections:*

**Reply**: This is a great point. We have organised the revised paper according to your recommendation. It does indeed make the paper more fluid to read.
* * *
*Q 8:* *Similarly, in Section 2.3, you could also explain that MBC is also applied to the blade-effective wind speed.*

**Reply**: We added the following to Section 2.4 (last paragraph):

*We have already mentioned that the measured inflow wind speeds were transformed to the non-rotating frame of reference by applying the MBC transformation. In order to assess the performance efficiency of the blade-mounted lidar-based inflow wind speed measurement, the blade effective wind speeds were also transferred into the non-rotating frame using the MBC transformation as follows*

$$
\begin{bmatrix} u_{bef,col} \\ u_{bef,yaw} \\ u_{bef,tilt} \end{bmatrix} = T_{mbc}(\theta) \begin{bmatrix} u_{bef,1} \\ u_{bef,2} \\ u_{bef,3} \end{bmatrix}
\tag{4}
$$

*where $T_{mbc}(\theta)$ is defined in Equation (9).*
* * *
*Q 9:* *The paragraph about the control development (page 9), the remarks, the $G_{d,f}^{-1}$ and the performance weight is not important for the rest of the paper and should be removed. Again, it seems to be an additional, not well explained and unnecessary complexity.*

**Reply**: We agreed that including $G_{d,f}^{-1}$ is not crucial for the paper and we have removed it in the revised manuscript. However, we think the the performance weight is required to show the objective in the control development.
* * *
*Q 10:* *Equation (6) and (7): Since the whole paper focus on the two sets of signals, Function f should be either explained in detail or simply avoided. Again, it seems to be an additional, not well explained and maybe not necessary complexity.*

**Reply**: We added the following sentences to the end of Section 2.2:

*The second-order polynomial function ( f ) is fitted on the data extracted from 10-minute large-eddy simulations with laminar inflow for mean wind speeds between $4\,m\,s^{-1}$ and $25\,m\,s^{-1}$. The $u(F, R)$ is the wind speed at an upstream distance of F from the blade, and at a blade radial position of R, and $u_0$ is taken from the same blade radial position of R, but at an upstream distance of three times the rotor diameter (3D).*
* * *
*Q 11:* *Section 3.1 explains the simulation setup using PALM, which then seems to be used in Section 3.3. In Section 3.2 however, generic wind speed measurements are used. It would be better in my opinion to switch them.*

**Reply**: Thank you for this suggestion. We have switched the sections in the revised manuscript.

**Minor issues**

*Q 12: Page 6, line 12: to estimate $u_{h,est,i}$ from Equation (1) to (3), you also need to neglect the weighing function. This is missing in the assumptions leading to Equation (4). Further, the expression "the measured LOS can be corrected" might be misleading, since the LOS are correct, you use Equation (4) to estimate or reconstruct the longitudinal wind speed.*

**Reply**: We updated this section by added the following to Section 2.2 first paragraph:

*Without loss of generality, the weighting function of $W(F,\xi)$ from Equation (1) was neglected, and two assumptions were made: (1) the $v_{h,i}$ and $w_{h,i}$ components are zero and (2) the mean wind speed is parallel with the rotor axis, i.e., no tilt and no yaw misalignments are considered.*

*Q 13: Several variables are introduced relatively late, e.g. $k$, $V_i(\xi)$.*

**Reply**: In the updated manuscript, we introduced the variables earlier in Section 2.1.

*Q 14: The variables are not consistently named: you use "blade-effective wind speed" for (1) the original $u_{bef,i}$ with $i$ for blade 1, 2, and 3, as well as (2) for the transformed $u_{bef,k}$, for $k \in \{col,yaw,tilt\}$.*

**Reply**:

As with many papers, there are many variables and we feel that it is sometimes more confusing to have different variable names for similar quantities. Thus we have used different indices $i$ and $k$ to distinguish between the different blade-effective wind speeds here.

*Q 15: Section 2.6: It is not clear that $u_{cor,k}$, is delayed. The only delay introduced in Section 2.3 is for the pitch angles.*

**Reply**: By applying the recommended method, this part has been removed from the updated manuscript.

*Q 16: The simulation time is not stated in Section 3.1, but might be interesting for all the frequency estimates. Sorry, if I missed that information somewhere else.*

**Reply**: Thank you for pointing this out, as we did indeed forget to specify the simulation time. We have corrected this in the revised manuscript and added the following to Section 3.2:

[revised manuscript text omitted]

10  We have already mentioned that the measured inflow wind speeds are transformed to the non-rotating frame of reference by applying the MBC transformation. In order to assess the performance efficiency of the blade-mounted lidar-based inflow wind speed measurement, the blade effective wind speeds are also transferred into the non-rotating frame using the MBC transformation as follows

$$
\begin{bmatrix} u_{\mathrm{bef,col}} \\ u_{\mathrm{bef,yaw}} \\ u_{\mathrm{bef,tilt}} \end{bmatrix} = T_{\mathrm{mbc}}(\theta) \begin{bmatrix} u_{\mathrm{bef,1}} \\ u_{\mathrm{bef,2}} \\ u_{\mathrm{bef,3}} \end{bmatrix}
\tag{10}
$$

where $T_{\mathrm{mbc}}(\theta)$ is defined in Equation (9).
* * *
[84]removed: was higher

[85]removed: was

[86]removed: were

[87]removed: Equation (8)

[88]removed: were

[89]removed: by considering $n_{\mathrm{h}}$ as 1 in Equation (8)

[90]removed: was

[91]removed: high

[92]removed: large

[Figure]

**Figure 5.** Block diagram of the disturbance rejection control design with performance weight and uncertain input measurement. $K_{\text{fb,f}}$, $K_{\text{ff,f}}$ are the feedback and feedforward controllers, $G_{\text{wt,f}}$ is the wind turbine model from the control input to output, $G_{\text{d,f}}$ is the wind turbine model from the disturbance to the output, $G_{\text{n,f}}$ is the nominal disturbance measurement model, $\Delta_\ell$ is the uncertainty, $W_\ell$ is the measurement uncertainty weight, and $W_{\text{p}}$ is the performance weight. The f in the index refers to the non-rotating (fixed) frame of reference.

**2.5   System modeling with uncertain lidar measurements**

We [..[93] ]use the blade-mounted telescopes to measure the disturbance, or the inflow wind speed in this case. Afterward, the three measurements [..[94] ]are transformed into the non-rotating frame of reference where they [..[95] ]are used as inputs to the feedforward individual and collective pitch controllers. Figure 5 illustrates the disturbance rejection controller setup with uncertainty. Each block in the figure represents a three-input and three-output system [..[96] ]with a $3 \times 3$ matrix [..[97] ]transfer function.

The control development [..[98] ]is aimed at achieving disturbance rejection up to a certain frequency with measurement uncertainties. In other words, we [..[99] ]want to find a controller that satisfies Equation (11) for a chosen performance weight $W_{\text{p}}$.

$$\left\| \, W_{\text{p}} \, S_{\text{fb}} \, S_{\text{ff,p}} \, \right\|_\infty < 1, \tag{11}$$

where the frequency-dependent feedback ($S_{\text{fb}}$) and feedforward sensitivity ($S_{\text{ff,p}}$) functions with [..[100] ]additive uncertainty are given by

$$
\begin{aligned}
S_{\text{fb}} &= \left(I + G_{\text{wt,f}} K_{\text{fb,f}}\right)^{-1}, \\
S_{\text{ff,p}} &= I + G_{\text{wt,f}} K_{\text{ff,f}} \left(G_{\text{n,f}} + \Delta_\ell W_\ell\right) G_{\text{d,f}}^{-1},
\end{aligned}
\tag{12}
$$
* * *
[93]removed: used

[94]removed: were

[95]removed: were

[96]removed: . Consequently, the resulting transfer function was in

[97]removed: (three-input and three-output). The measurement uncertainty can vary with wind speed, wind shear, turbulence intensity, etc. (**?**), thus, multi-plicative diagonal complex uncertainties were considered.

[98]removed: was

[99]removed: wanted

[100]removed: multiplicative

10 and

$$\Delta_\ell = \begin{bmatrix} \delta_{\ell,1} & 0 & 0 \\ 0 & \delta_{\ell,2} & 0 \\ 0 & 0 & \delta_{\ell,3} \end{bmatrix} \in \mathbb{C}^{3\times3} \ , \ ||\Delta_\ell||_\infty \leq 1. \tag{13}$$

[..[101] ]This equation highlights the importance of knowing the frequency-dependent uncertainty weight $W_\ell(j\omega)$ in advance, so as to ensure that the closed-loop system is stable and that the objective in Equation (11) is satisfied for all perturbations ($||\Delta_\ell||_\infty \leq 1$). For control development, [..[102] ]the frequency dependent uncertainty weight of $W_\ell(j\omega)$ [..[103] ]and the nominal disturbance measurement model of $G_{n,f}(j\omega)$ are missing, we discuss how they can be identified in the next subsections and later illustrate the process for the reference [..[104] ]cases in Section 3.3.

5 *[..[105] ]*Remark*: [..[106] ]*Only one objective [..[107] ]is introduced in Equation (11); nevertheless, other objectives can be added, such as penalizing the control signal magnitude at high frequencies (Ungurán et al. (2019)). [..[108] ]

[..[109]]

[..[110] ]

[..[111]]

10 [..[112] ]

**2.6 Uncertainty modeling for control development**

We [..[113] ]employ black box system identification to establish the transfer functions ($G_\ell$) from the blade effective wind speeds ($u_{beff}$) to the corrected [..[114] ]lidar-based inflow wind speeds ($u_{cor}$) in the non-rotating (fixed) frame of reference

$$u_{cor,f} = G_\ell u_{beff,f} \tag{14}$$

15 with

$$G_\ell = \begin{bmatrix} G_{\ell,col} & 0 & 0 \\ 0 & G_{\ell,yaw} & 0 \\ 0 & 0 & G_{\ell,tilt} \end{bmatrix} \in \mathbb{C}^{3\times3} \ . \tag{15}$$
* * *
[101] removed: for property $||\Delta_\ell||_\infty \leq 1$.

[102] removed: only the identification of

[103] removed: was missing, which was identified

[104] removed: case

[105] removed: Remarks

[106] removed: (1)

[107] removed: was

[108] removed: (2) To avoid the disturbance model acting as a scaling factor of the objective function, as in

[110] removed: Figure 5 was extended with the inverse of the disturbance model ($G_{d,f}^{-1}$) (shown in a dashed rectangle), so that

[112] removed: which ensures that $z_1$ is not affected by the disturbance model. Hence, in the control synthesis and analysis, $z_1$ is a direct indicator of the controller performance in the presence of uncertainties.

[113] removed: employed

[114] removed: lidar based

The system identification is performed via the `ssest` function from MATLAB (2018) with a 15th-order state-space model, which can capture all the relevant information. The order of the state-space model [..[115] ]is found empirically through analysis of the Hankel singular values.

We separately [..[116] ]identify the nominal disturbance measurement model ($G_{\mathrm{n},k}(j\,\omega)$) and the [..[117] ]

[..[118]]

[..[119] ]

[..[120]]

5   uncertainty weight ($w_{\ell,k}(j\,\omega)$), where

[..[121]]

[..[122] ]$k \in \{\mathrm{col, yaw, tilt}\}$, as a 5th-order minimum phase filter for each of the inputs in such a way as to satisfy the following inequalities

$$|G_{\mathrm{n},k}(j\,\omega)| \quad < \quad |G_{\ell,k}(j\,\omega)|,\ \forall\omega, \tag{16}$$

10   and

$$|G_{\mathrm{n},k}(j\,\omega) + w_{\ell,k}(j\,\omega)| \quad > \quad |G_{\ell,k}(j\,\omega)|,\ \forall\omega, \tag{17}$$

leading to the [..[123] ]diagonal nominal disturbance measurement model matrix of

$$G_{\mathrm{n}} = \begin{bmatrix} G_{\mathrm{n,col}} & 0 & 0 \\ 0 & G_{\mathrm{n,yaw}} & 0 \\ 0 & 0 & G_{\mathrm{n,tilt}} \end{bmatrix}, \tag{18}$$

[..[124] ]and [..[125] ]uncertainty weight matrix of

$$\qquad W_{\ell} = \begin{bmatrix} w_{\ell,\mathrm{col}} & 0 & 0 \\ 0 & w_{\ell,\mathrm{yaw}} & 0 \\ 0 & 0 & w_{\ell,\mathrm{tilt}} \end{bmatrix}. \tag{19}$$
* * *
[115]removed: was found empirically

[revised manuscript text omitted]

* * *
[153] removed: aside from
[154] removed: was
[155] removed: harmonics
[156] removed: rose
[157] removed: was
[158] removed: resulted to
[159] removed: Uncertainty weight identification
[166] removed: stressed
[167] removed: was
[168] removed: the

**Table 1.** The cases investigated in this study, along with the lidar and telescope parameters for each case. If one or more parameters in the third column are not specified, then the parameters defined in the first case are used. $F$ is the focus length, $R$ is the radial position of the telescope along the blade, and $\Phi_{\ell,i}$ and $\Gamma_{\ell,i}$ are the orientation angles of the telescope (see Figure 2).

| Case | [..160 ]Conditions | Parameters |
|---|---|---|
| $C_1$ | telescope parameters from [..161 ]literature, assuming:
 – no induction
 – no wind evolution
 – no blade flexibility
 – constant rotor speed
 – constant blade pitch angles | $F = 22.2\,\mathrm{m}$
 $R = 44\,\mathrm{m}$
 $\Phi_{\ell,i} = -3.7°$
 $\Gamma_{\ell,i} = 7.0°$ |
| $C_2$ | telescope parameters within prescribed range | $F \in [20.2\,\mathrm{m}, 30\,\mathrm{m}]$
 [..162 ]$R \in [42\,\mathrm{m}, 47\,\mathrm{m}]$
 [..163 ]$\Phi_{\ell,i} \in [-6.7°, -0.7°]$
 [..164 ]$\Gamma_{\ell,i} \in [4°, 10°]$ |
| $C_3$ | different telescope focus length | $F \in [20.2\,\mathrm{m}, 30\,\mathrm{m}]$ |
| $C_4$ | different position of the telescope along the blade radius | [..165 ]$R \in [42\,\mathrm{m}, 47\,\mathrm{m}]$ |
| $C_5$ | different orientation angles of the telescope | $\Phi_{\ell,i} \in [-6.7°, -0.7°]$
 $\Gamma_{\ell,i} \in [4°, 10°]$ |
| $C_6$ | telescope orientation misalignment | $\Phi_{\ell,i} = \Phi_{\ell,1} \pm 5°$
 $\Gamma_{\ell,i} = \Gamma_{\ell,1} \pm 5°$
 with $i = 2, 3$ |

sensitivity function may increase due to uncertainties in the system. Therefore, it is important to [..[169] ]analyse how the lidar measurement uncertainty is affected by e.g., mounting misalignment of the telescope on the blade, or in cases where the focus distance or position of the telescope along the blade span differs from the optimal parameters, etc. Identifying the lidar measurement uncertainty as a frequency-dependent [..[170] ]minimum-phase filter enables the inclusion of such parameters in the control development, allowing an analysis of its impact on the stability and performance of the closed-loop system. [..[171] ]As we explain in details in Section 3.3.1, a straightforward solution to determine the telescope and lidar parameters, such as focus distance, telescope position along the blade radius, telescope orientation on the blade, etc., is to assume that the blades are rigid, that the rotor speed and pitch angle are constant, and that Taylor's frozen turbulence hypothesis (Taylor (1938)) holds (Ungurán et al. (2018)). We [..[172] ]perform large-eddy simulation (LES) in the succeeding sections to examine the usefulness and limitations of these assumptions, and further [..[173] ]analyse the uncertainties in the blade-mounted lidar measurement as well as the measurement sensitivity with respect to lidar and telescope parameter changes. The investigated cases are described in Sections 3.3.1 to 3.3.5 [..[174] ]and summarized in Table 1. Section 3.3.6 describes how the measurement uncertainties are affected when one or two telescopes are aligned differently than the others. First, we [..[175] ]assume that the orientation angle misalignment [..[176] ]is unknown. Second, we [..[177] ]assume that this orientation angle misalignment can be identified, so that the lidar-based inflow wind speed measurement can be corrected. [..[178] ]

For each case, [..[184] ]first the transfer functions ($G_{\ell,k}$) from the blade effective wind speeds ($u_{\text{beff},k}$) to the corrected lidar-based inflow wind speeds ($u_{\text{cor},k}$) are identified. Next, the [..[185] ]nominal disturbance measurement models ($G_{\text{n},k}$) and the uncertainty weights ($w_{\ell,k}$) for each of the inputs are estimated to satisfy [..[186] ]Equations (16) and (17). Figure 10 provides a summary of the [..[187] ]identified DC gain upper ($G_{\text{n},k} + w_{\ell,k}$) and lower ($G_{\text{n},k}$) bounds of the transfer functions ($G_{\ell,k}$) from the blade effective wind speeds ([..[188] ]$u_{\text{beff},k}$) to the corrected lidar-based inflow wind speeds ([..[189] ]$u_{\text{cor},k}$).

We would like to act only below the 1P (0.195 [..[190] ]Hz) frequency, therefore, below this frequency, it is desired that the gain of $G_{\text{n},k}$ is 1, and that the measurement uncertainty is small, but still covers the worst case. A higher percentage of measurement uncertainty can be tolerated at frequencies above 1P by designing the feedforward controller accordingly,
* * *
[169]removed: analyze

[170]removed: first-order

[171]removed: A

[172]removed: performed

[173]removed: analyzed

[174]removed: ,

[175]removed: assumed

[176]removed: was

[177]removed: assumed

[178]removed: were

[184]removed: the relative error between the nominal ($G_{\text{n},k}(j\omega)$) and the identified ($G_{\ell,k}(j\omega)$) systems were first determined

[185]removed: uncertainty weight parameters from Equation (17) were

[186]removed: **??**

[187]removed: estimated parameters. The DC ($w_{\text{DC},k}$) and high-frequency gains ($w_{\infty,k}$) of the filter were expressed in percentage, representing the normalized system perturbation away from 1 on that frequency. Thus, 0 % of uncertainty indicates that the identified transfer function ($G_\ell$

[188]removed: $u_{\text{beff}}$

[189]removed: $u_{\text{cor}}$)can have a gain of 1 in that frequency. Moreover, 10

[190]removed: % of uncertainty means that the identified transfer function ($G_\ell$) can have a gain of either 0.9 or 1.1 in that frequency.

[Figure]

**Figure 10.** Identified [..[179] ]DC gain upper ($G_{n,k} + w_{\ell,k}$) and [..[180] ]lower ($G_{n,k}$) bounds of the [..[181] ]transfer functions ($G_{\ell,k}$) from the blade effective wind speeds ($u_{beff,k}$) to the corrected lidar-based inflow wind speeds ($u_{cor,k}$), with $k \in \{col, yaw, tilt\}$; $C_1$, $C_2$, $C_3$, $C_4$, and $C_5$ represent the investigated cases ([..[182] ]outlined in [..[183] ]Table 1).

[revised manuscript text omitted]